# Quantum Gases of Dipoles, Quadrupoles and Octupoles in Gross–Pitaevskii Formalism with Form Factor

**Artem A. Alexandrov, Alina U. Badamshina and Stanislav L. Ogarkov \*** 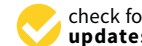

Moscow Institute of Physics and Technology (MIPT), Institutskiy Pereulok 9,
141701 Dolgoprudny, Moscow Region, Russia; aleksandrov.aa@phystech.edu (A.A.A.);
badamshina.au@phystech.edu (A.U.B.)

**\*** Correspondence: ogarkovstas@mail.ru

**Abstract:** Here, classical and quantum field theory of dipolar, axisymmetric quadrupolar and octupolar Bose gases is considered within a general approach. Dipole, axisymmetric quadrupole and octupole interaction potentials in the momentum representation are calculated. These results clearly demonstrate attraction and repulsion areas in corresponding gases. Then the Gross–Pitaevskii (GP) equation, which plays a key role in the present paper, is derived from the corresponding functional. The zoology of the form factors appearing in the GP equation is studied in details. The proper classes for the description of spatially non-uniform condensates form factors are chosen. In the Thomas–Fermi approximation a general solution of the GP equation with a quasilocal form factor is obtained. This solution has an interesting form in terms of a double rapidly converging series that universally includes all the interactions considered. Plots of condensate density functions for the exponential-trigonometric form factor are given. For the sake of completeness, in this paper we consider the GP equation with an optical lattice potential in the limit of small condensate densities. This limit does not distinguish between dipolar, quadrupolar and octupolar gases. An important analysis of the condensate stability, in other words the study of condensate excitations, is also performed in this paper. In the Gaussian approximation (from the Gross–Pitaevskii functional), a functional describing the perturbations of the condensate is derived in detail. This problem is an analog of the Bogolubov transformation used in the study of quantum Bose gases in operator formalism. For a probe wave function in the form of a plane wave, a spectrum of (Bogoliubov) excitations was obtained, from which an equation describing the threshold momentum for the emergence of instability was derived. An important result of this paper is the dependence of the threshold on the momentum of a stationary condensate. For completeness of the presentation, the approximating expression in the form of a rapidly converging series is obtained for the corresponding dependence, and plots of the corresponding series for the exponential-trigonometric form factor are given. Finally, in the conclusion a quantum hydrodynamic theory for dipolar, axisymmetric quadrupolar and octupolar gases is briefly presented, giving a clue to the experimental determination of the form factors.

**Keywords:** quantum field theory (QFT); scalar QFT; nonlocal QFT; Euclidean QFT; quantum gases; dipoles; quadrupoles; octupoles; Gross–Pitaevski (GP) formalism; form factor

## 1. Introduction

Discussing the non-relativistic systems of many particles, we are faced with a colossally complex problem of solving the Schrödinger equation for an enormous number of particles. To date,

standard methods for solving this problem are the methods of quantum field theory (see, for example, the monographs [1–9]): in terms of the functional integral over theory's primary fields, a generating functional of Green functions or $n$-particle ($n \geq 1$) irreducible vertices (the corresponding functional is called an effective action) is formulated. Further calculation of this functional integral is carried out in terms of perturbation theory, which leads to the construction of a diagram technique (Feynman diagrams) for generating functionals, or immediately for the corresponding families of Green functions.

Since the expressions constructed using the diagram technique are asymptotic series, the next step in the obtaining of a meaningful answer is the application of summation methods for asymptotic series, for example, the Borel–Laplace method. However, the latter is more common among the quantum field community [6–9]. In a solid-state physics community, we usually try to sum the subsequences of the diagrams in such a way that the final expressions for the Green functions make sense, in other words, try to find a "re-expansion" with respect to some new effective coupling constant [1–5].

Another method of calculating functional integrals is the saddle-point method. Within this method an effective (for example, already taking into account a series of diagrams of a certain type, or various nonperturbative effects, not "captured" by perturbation theory) equation of motion for a certain Green function (the saddle-point method is well described in [7]). An example of such an equation is the well-known Gross–Pitaevskii (GP) equation, repeatedly applied for description of quantum gases of bosons, fermions and their mixtures [10,11]. Thus, the GP equation has a fundamental microscopic meaning.

At the same time, we can find ourselves in the following situation: The saddle-point method may not "catch" the required behaviour of the quantum gas. Simultaneously, this behaviour may not be achieved within the diagram technique. These situations are common in literature, and one of the generally accepted methods is the functional (nonperturbative, exact) renormalization group method [8,9,12]. Alternatively, we can try to develop a semi-phenomenological model. For instance, the simplified (stochastic) quantum field model is widely used for the fundamental turbulence phenomenon description. Taking into account the experience of the semi-phenomenological description, a model based on the GP equation with a phenomenological form factor is proposed in the present paper (the analysis of the form factors in the context of quantum field theory is performed in [13,14]. What is known about such a "building block of the theory" from the most general considerations? It should correctly reproduce different distributions of several reference physical quantities in the system and give a qualitative picture. Therefore, there are no univocal rules for choosing this building block (the ambiguity in the choice can be reduced by restrictions due to the renormalization group, the Ward identities or the Schwinger–Dyson equations). In defence of such a vulnerable for criticism state, a number of generally accepted arguments is presented (see [7]).

In microscopic theories, these building blocks must be generated by various microscopic mechanisms, and their characteristics for a particular problem must be computable. However, if there is no microscopic theory of this kind, within an effective description, which is only a simplified semi-phenomenological version of the (hypothetical) accurate theory, a specific choice of the form factor can be justified by general considerations and results.

Further, let us provide a brief literature review reflecting on a current situation. Here it should be noted that any choice is subjective hence, by definition, incomplete. For example, it can be said that the quantum Bose gases science was revitalized because of the relatively recent series of experimental papers [15–17] on the direct observation of the Bose–Einstein condensation phenomenon in various atomic gases in traps. A number of theoretical publications in this field has significantly increased since those experimental successes. Only in the last decade studies such as Bose–Einstein condensation in dipole systems [18,19], different modes of electron-hole pairing in certain graphene-based structures, in particular in graphene bilayer [20–22], and similar pairing in very popular to date topological insulators thin films [23] can be noted. Both theoretical and experimental situations are described in detail in the remarkable review [24]. Here we should also mention a very trustworthy theoretical

work about exciton-roton excitations in two-dimensional Bose gases of quadrupoles [25]. It should be noted, that in this paper the transition from the three-dimensional case to the two-dimensional case is performed strictly and consistently.

Talking about the phenomenon of Bose–Einstein condensation (BEC), let us note a brief description of the possibility of obtaining a Bose–Einstein condensate from a more general phenomenon—the BCS-BEC crossover phenomenon. This phenomenon, valid in cold quantum Fermi gases, consists in changing of the behaviour of the corresponding gas upon the transition from superconducting Bardeen–Cooper–Schrieffer (BCS) pairing to Bose–Einstein condensate of molecules strongly localized in the coordinate space. Due to that we note reviews [26,27], and also an interesting review [28] devoted to a wide range of condensed matter physics phenomena using the example of ultracold atomic gases in optical traps. Finally, a very intriguing theoretical approach to describing various condensed matter physics phenomena is the so-called "holographic" approach, originated from fundamental papers [29–31]. A lot of publications are focused on this approach, among which we note papers [32,33], focused on holographic picture for d-wave superconductor. This concludes the analysis of the literature and we proceed to the description of the structure of our paper.

This paper consists of five main sections and one appendix. After the introduction there is "Multipole interactions" section, where classical interaction potentials in dipolar, axisymmetric quadrupolar and octupolar gases are calculated in coordinate $r$ representation as well as in momentum $k$ representation, also the expression for the $l$-pole momentum interaction is presented. We note that various modifications of this interaction are also discussed in our paper. In addition to the classical modification with core (a hard core illustrating a hard repulsion at short distances) an original modification with "split" is proposed. This mechanism implies the splitting of the interaction by a certain feature, for example, by the sign of the initial interaction range.

The next section, "The Gross–Pitaevskii equation", is central to this paper. It starts with obtaining of the Gross–Pitaevskii equation, which models the interaction action as an effective result of calculation of the functional integral. The GP equation is derived both in the coordinate and the momentum representation. The key feature of this model is that it allows considering almost a general scenario of the spatially non-uniform condensate formation and its excitations (of an arbitrary origin). This semi-phenomenological model allows describing the physics of practically any quasiparticles arising in considered quantum Bose gases.

Then follows a detailed study of various types of form factors. It was considered that for a wide class of physical scenarios the so-called "quasilocal" form factors are sufficient. A general solution of the GP equation is obtained for this type of form factors in Thomas–Fermi approximation. This solution is one of the main results of this paper. This solution is presented in terms of a double rapidly converging series (the last statement can be proved directly by constructing the corresponding majorant). In addition to the above the obtained solution has a universal form in terms of the dimension of the space $D$ (for specific calculations we use $D = 3$). The obtained solution also universally includes all types of Bose gases considered.

The results are demonstrated in the form of plots of condensate density functions for the exponential-trigonometric form factor, because the exponent has good properties for numerical calculations, and the trigonometric argument is perfect for reproducing of a stationary periodic structure in the system. For the sake of completeness the GP equation in the limit of small condensate densities is also considered, together with the case when the Bose gas is in a trap. The potential is chosen in a form of an optical lattice. The limit of small condensate densities does not distinguish between dipolar, quadrupolar and octupolar gases, because the non-linear term is equal to zero in this limit. The linear limit of the GP equation with a potential of an optical lattice is a single-particle Schrödinger equation in a periodic potential, and its stationary analogue is the Mathieu equation. Analytical results for the energy spectrum of this quantum mechanical system are presented in this paper in the limit of weak coupling as well as strong coupling in terms of the parameters of the trap (with reference to [34,35]).



The study of the properties of the stationary solution of the GP equation cannot be considered complete unless the analysis of the stability of the stationary solutions is made. For this reason an important analysis of the condensate stability is performed in the section "The analysis of the condensate excitation". In Gaussian approximation the functional, which is an action for different condensate perturbations, is derived in details from the GP functional. We only consider the case of a quasilocal form factor of a special form for this problem (the amplitude of the form factor depends only on the space coordinate $r$). The solved problem is alternative to the problem of the Bogoliubov transformations used in the studies of the quantum Bose gases in the operator formalism. At the same time, the considered case is an essential generalization of a standard one. The generalizations are the presence of a structure in the system, as well as the arbitrariness of the probe wave function of the condensate perturbations. To obtain the Bogoliubov spectrum, this wave function was chosen in the form of a plane wave.

As the next step, an equation describing the threshold (critical) perturbation momentum was obtained from the Bogoliubov spectrum, at which instabilities arise. This equation is also one of the main results of the paper. The equation has an original form, since it involves sufficiently general form of the theory's form factor. At the same time, various dependencies between the characteristics of the condensate and its excitations can be studied with help of this equation. Unlike the simplest problems focused on the Bogoliubov transformation such an equation includes realistic scenarios of the interactions between two subsystems. In particular, it allows us to determine the dependence of the threshold of the condensate excitation from the specific momentum of the condensate. An approximating formula for this dependence is also derived in this paper. The approximating formula has a form of a rapidly converging series and also a universal form in terms of the dimension of the space $D$ (in specific calculations we again only use three-dimensional Bose gases, corresponding to the interactions which we derived in the classical theory) and the types of the gases considered. For the specified dependence of the instability threshold from the specific momentum of the condensate the corresponding plots are constructed for the exponential-trigonometric form factor.

The Section 5 sums up the results, possible further directions in the study of the quantum Bose gases in a model with a form factor, and also a question about the experimental determination of the form factor is discussed. To answer this question the hydrodynamics of the quantum Bose gases of dipoles, quadrupoles and octupoles is briefly considered (in the spirit of the monograph [10]). It follows from hydrodynamic theory that a form factor can be determined experimentally because it directly determines the corresponding generalized Euler equation. Moreover, the obtained system of equations of hydrodynamics is an alternative approach for determination of Bogoliubov spectrum for an excited system which naturally corresponds with the known analysis of the system's stability. We note that an important consequence of hydrodynamics is the fact that quasilocal form factors turn out to be the most "natural" ones: more complex physical mechanisms must be turned on for the emergence of the non-locality of the generalized Euler equation. For this reason the phenomenology of the model proposed in this paper is minimal. Thus, the proposed description of quantum Bose gases in terms of the Gross–Pitaevskii equation with a quasilocal form factor is interesting both from the theoretical point of view and for practical applications such as experiments with Bose gases of dipoles, axisymmetric quadrupoles and octupoles. In the Appendix A of this paper focused on the classical part of the problem of describing Bose gases detailed derivations of the interaction potentials of classical gases with different values of multipolarity both in the coordinate $r$ and the momentum $k$ representations.

According to the review [24], the dipole–dipole (drawing parallels: quadrupole-quadrupole and octupole-octupole) interaction, acting between particles having a permanent electric or magnetic dipole (quadrupole and octupole, respectively) moment, should lead to a novel kind of degenerate quantum gas already in the weakly interacting limit. Its effects should be even more pronounced in the strongly correlated regime. Candidates for the role of quantum gases with dipole interaction are given in Section 1.2 "Interactions" of the review [24]. The properties of the dipole–dipole interaction are

radically different from the ones of the contact interaction. The dipole–dipole interaction is long-range and anisotropic which leads to completely new physical phenomena (see for example the case of ferrofluids). Similarly, anisotropy of interactions lies behind the fascinating physics of liquid crystals. All this is argued in the review [24].

The parameters of dipole gases are given in the Table 1 "Dipolar Constants for Various Atomic and Molecular Species" in Section 3 "Creation of a Dipolar Gas" in the review [24] as well as an overview of candidates (polar molecules, Rydberg atoms, light-induced dipoles, magnetic dipoles). Detailed experimental implementations are presented. In our paper, the expression (20), we give the relative amplitudes of the interaction in dipole, quadrupole and octupole gases. For this reason, the experimental parameters of gases can be predicted.

The the Section 4 "Non-Local Gross–Pitaevskii Equation" of the review [24] an equation with non-local term is given (due to the long-range character of the dipolar interaction). This term makes it much more complicated to solve the Gross–Pitaevskii equation, even numerically, as one now faces an integro-differential equation. The experimental realisations for validation of this term are discussed. Various versions of the pseudopotential, which must be substituted into the equation instead of the bare interaction, are also discussed, as well as the hydrodynamics derived from this equation. The form factor is an analogue of the pseudopotential, so the results of the review [24] remain valid in our paper.

Further, there is the discussion of trapped gases (general theoretical model including quadrupoles and octupoles is presented in our paper) and quantum ferrofluid as well as physical realizations of the latter in the review [24]. Such realizations, in principle, can be established for quadrupoles and octupoles with an appropriate level of experimental capabilities. Moreover, dipolar gases in optical lattices are also reviewed in [24]. The generalization of the theory to quadrupole and octupole gases is the main task of our paper. Experimental implementation of this remains a matter of the future.

Finally, let us note that the quantum gases with quadrupole-quadrupole interactions are widely discussed [25]. One can formulate essential properties of the quantum gas constituents as it was done in [36] in case of Fermi system with explicit example of Yb and Sr atoms. Then, the question about quadrupole-quadrupole and octupole-octupole interactions is interesting in the context of exotic phases in ultra-cold Bose systems.

## 2. Multipole Interactions

We begin with the discussion of multipole interactions. In doing so we will consider systems of identical multipoles with axial symmetry which simplifies the calculations. Expressions for multipolar interactions will be the necessary "building blocks" in the problem of the Gross–Pitaevskii equation with a form factor for ultracold Bose gases of a various multipolarity. Detailed calculations related to this section are given in the Appendix A.

### 2.1. D-D Interaction

In the coordinate representation, the expression for the energy of the dipole-dipole interaction is well-known and has the form of:

$$U_D(\boldsymbol{r}, \psi) = \frac{d^2(1 - 3\cos^2\psi)}{r^3} = \frac{-2d^2}{r^3}P_2(\cos\psi), \tag{1}$$

where $\psi$ is the angle between the axis of symmetry of a dipole and the vector $\boldsymbol{r}$. Next the Fourier transform has to be done, but before that we need to transform from the angle $\psi$ to the angle $\alpha$ between the vector $\boldsymbol{k}$ and the axis of symmetry of the dipole. Here and further we denote the $n$-th Legendre polynomial by $P_n$. Without loss of generality and to simplify the calculations the axis of symmetry can be considered as laying on the $y = 0$ plane, then the equation which relates the angles $\psi$ and $\alpha$ can be written in the form:

$$\cos\psi = \sin\alpha\cos\varphi\sin\theta + \cos\alpha\cos\theta. \tag{2}$$

In the momentum representation the expression for the dipole-dipole interaction energy is written in the form [24]:

$$U_D(\mathbf{k}, \alpha) = \frac{4\pi d^2}{3}(3\cos^2\alpha - 1) \equiv A_D P_2(\cos\alpha), \tag{3}$$

where $\alpha$ is the angle between the direction of the dipole moment $\mathbf{d}$ and the vector $\mathbf{k}$ and $A_D = 8\pi d^2/3$. It should be noted that in the momentum representation the D-D interaction does not depend on the absolute value of the vector $\mathbf{k}$. Now let us consider the Q-Q interaction.

### 2.2. Q-Q Interaction

Calculating the energy of the Q-Q interaction in the $\mathbf{r}$-representation is a simple problem. Consider the interaction between two quadrupoles with equal quadrupole moment $Q_{\mu\mu}$. Tensor $Q_{\mu\mu}$ has a form of:

$$Q_{\mu\nu} = \text{diag}\left(-\frac{Q}{2}, -\frac{Q}{2}, Q\right). \tag{4}$$

Here $Q = \int (2z^2 - x^2 - y^2)/2\,dq$. The potential and the interaction energy have a form of

$$\varphi = \frac{Q_{\alpha\beta}x_\alpha x_\beta}{2r^5}; \quad U_Q(\mathbf{r}) = \frac{Q_{\mu\nu}}{6}\partial_\mu\partial_\nu\varphi. \tag{5}$$

Performing the calculations taking into account the connection between the angles $\theta$ and $\psi$ we obtain the following equation:

$$U_Q(\mathbf{r}, \psi) = \frac{3Q^2}{16r^5}\left(3 - 30\cos\psi^2 + 35\cos^4\psi\right). \tag{6}$$

Or in a more compact way:

$$U_Q(\mathbf{r}, \psi) = \frac{3Q^2}{2r^5}P_4(\cos\psi). \tag{7}$$

Then in the momentum representation we have:

$$U_Q(\mathbf{k}, \alpha) = \int d^3\mathbf{r}\, U(\mathbf{r}, \psi)e^{-i\mathbf{k}\mathbf{r}}, \tag{8}$$

where $\alpha$ is the angle between the axis of symmetry of the quadrupole and the vector $\mathbf{k}$. After integrating the equation for the energy will be in the form of:

$$U_Q(\mathbf{k}, \alpha) = \frac{3Q^2\pi k^2}{16}\left(\frac{4}{35} - \frac{8}{7}\cos^2\alpha + \frac{4}{3}\cos^4\alpha\right). \tag{9}$$

The last expression can be written in a more compact and clear way:

$$U_Q(\mathbf{k}, \alpha) = A_Q k^2 P_4(\cos\alpha); \quad A_Q = \frac{2\pi Q^2}{35}. \tag{10}$$

It is clear that unlike the D-D interaction a dependence from the squared absolute value of the momentum $\mathbf{k}$ occurs.

### 2.3. O-O Interaction

The octupole moment tensor has a form of:

$$O_{\alpha\beta\gamma} = \int dq\left(15r_\alpha r_\beta r_\gamma - 3\delta_{\alpha\beta}r^2 r_\gamma - 3\delta_{\beta\gamma}r^2 r_\alpha - \delta_{\gamma\alpha}r^2 r_\beta\right). \tag{11}$$

For the octupole moment tensor in case of axial symmetry the following equations occur:

$$O_{\alpha\beta\gamma} = O_{\beta\alpha\gamma} = O_{\alpha\gamma\beta} = O_{\gamma\beta\alpha}. \tag{12}$$

In addition since $O_{\gamma\gamma\alpha} = 0$ the following equations hold:

$$O_{xxz} = O_{xzx} = O_{zxx} = O_{yyz} = O_{yzy} = O_{zyy} = -\frac{O_{zzz}}{2}. \tag{13}$$

Here and further $O_{zzz} = O$. In what follows, we denote $O_{zzz} = O$. The calculation of the interaction energy in the coordinate representation is a cumbersome but easy task. The interaction potential is given by:

$$U_O(\boldsymbol{r}, \psi) = \frac{O_{\alpha\beta\gamma}O_{\lambda\mu\nu}}{540}\partial_\alpha\partial_\beta\partial_\gamma\left(\frac{r_\lambda r_\nu r_\mu}{r^4}\right). \tag{14}$$

After calculations the following equation is obtained:

$$U_O(\boldsymbol{r}, \psi) = -\frac{5O^2}{9r^7}P_6(\cos\psi). \tag{15}$$

For detailed calculations see the Appendix A. In momentum representation the interaction reads as follows:

$$U_O(\boldsymbol{k}, \alpha) = A_O k^4 P_6(\cos\alpha); \quad A_O = \frac{4\pi O^2}{18711}. \tag{16}$$

In case of octupoles the potential energy depends on the forth degree of the absolute value of the momentum $\boldsymbol{k}$. The generalization for higher-order multipoles is obvious. Graphical illustrations of the interaction energy are shown in Figures 1 and 2.

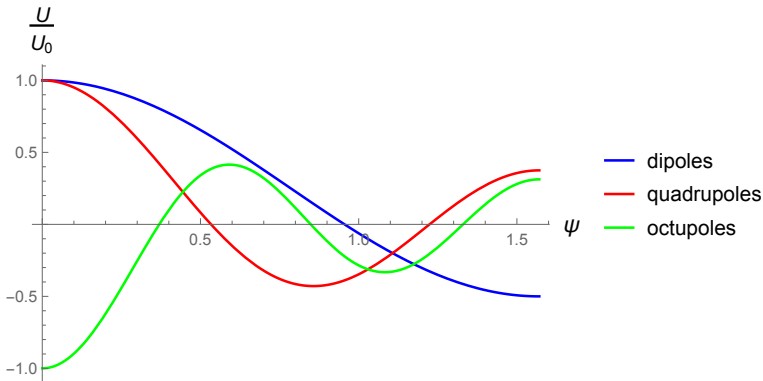

**Figure 1.** Normalized interaction energy in $\boldsymbol{r}$-representation.

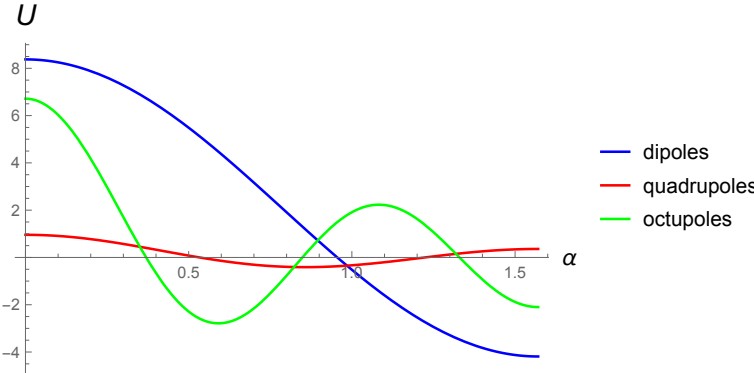

**Figure 2.** Interaction energy as a function of angles with a fixed momentum. For illustrative purposes the momentum in case of $U_O$ 10 times larger than the momentum $U_D$, $U_Q$.

### 2.4. Comparative Analysis

#### 2.4.1. Interaction Energy of Multipoles in General Terms

Using the properties of Legendre polynomials one can easily obtain the following equation:

$$P_l(\cos\psi) = \frac{U_l(\mathbf{r}, \psi)}{U_l(\mathbf{r}, 0)}. \tag{17}$$

The interaction energy as a function of the angle in $\mathbf{r}$-representation and in $\mathbf{k}$-representation is shown in Figures 1 and 2. The tensor of the $2^l$-pole moment in the axially symmetric case has a $2l+1$ non-zero component, $2l$ of which are equal to $-M/2$ and $(2l+1)$ are equal to $M$. Then the "amplitude" of the interaction, which represents the moment tensor contraction with itself, is simply the sum of the squares of the components, i.e., in the case of the $2^l$-pole moment, the interaction amplitude is:

$$2l\left(-\frac{M}{2}\right)^2 + M^2 = M^2\left(\frac{l}{2}+1\right). \tag{18}$$

For the interaction of two axially symmetric multipoles the following expression can be written:

$$U_M(\mathbf{r}, \psi) = K(l)\frac{M^2}{r^{2l+1}}\left(\frac{l}{2}+1\right)P_{2l}(\cos\psi), \tag{19}$$

where $K(l)$ is a function of the multipole order. Now let us turn to discussion of the results obtained.

#### 2.4.2. Discussion of Interactions in Momentum Representation

The results of calculations of the interaction energy in the $\mathbf{k}$-representation lead to the conclusion that starting from the Q-Q interaction the potential energy is comparable to the kinetic energy of the particles. In the case of the O-O interaction, the potential energy completely suppresses the kinetic energy at large momenta. It should also be noted that the amplitude of the interaction highly decreases with the increase of the multipole order. For example, for $d = Q = O$, one gets:

$$A_D \ : \ A_Q \ : \ A_O \approx 12500 : 270 : 1. \tag{20}$$

Here we note that these formulas are useful when one wants to investigate the scaling properties of a quantum Bose gas from general considerations, which is important not only for the Gross–Pitaevskii formalism, but also, for example, for the method of the functional renormalization group (with application to the quantum Bose gas).

This concludes our discussion of the classical theory of dipolar, quadrupolar and octupolar gases, and we turn to the central section of this paper focused on the Gross–Pitaevskii equation with a form factor (GP equation is widely discussed in the book [10]).

### 3. The Gross–Pitaevskii Equation

The description of the physics of various excitations of quantum Bose gases with a so-called "form factor" is key to this paper. This idea largely repeats the considerations made when one wants to describe a physical phenomenon using the apparatus of stochastic partial differential equations (SPDEs): we offer a semi-phenomenology in which the form factor (in the SPDEs theory this is the so-called "pump function" which is the Fourier transform of the correlator of a random variable) is chosen based on general considerations and is not derived from the microscopic theory.

What is known about such "building blocks" of theory as a form factor or a pump function from the most general considerations? These blocks must correctly reproduce the various distributions of energy, density and a number of other standard physical quantities in the system, give a qualitative picture (whether different wave structures are formed in the system, or there is stochasticity

semi-phenomenologically modeled by random force for example). Thus, there are no univocal rules for choosing this building block (the arbitrariness in the choice can be reduced by restrictions due to the renormalization group). In defence of such a vulnerable for criticism state, a number of generally accepted arguments is presented.

In microscopic theories, these building blocks must be generated by various microscopic mechanisms, and their characteristics for a particular problem must be computable. However, if there is no microscopic theory of this kind, within an effective description, which is only a simplified semi-phenomenological version of the (hypothetical) accurate theory, a specific choice of the form factor or the pump function can be justified by general considerations and results.

### 3.1. Equations in Real-Space and Momentum-Space Representations

The first principle in quantum field theory and statistical physics of quantum gases is the statement of the partition function of the theory in terms of the functional integral [6–9]:

$$\mathcal{Z} = \int \mathcal{D}\left[\bar{\psi}, \psi\right] e^{-S[\bar{\psi},\psi]}. \tag{21}$$

The action of the model consists of the action of the free theory $S_0$ and the action of interaction $S_1$:

$$S\left[\bar{\psi}, \psi\right] = S_0\left[\bar{\psi}, \psi\right] + S_1\left[\bar{\psi}, \psi\right]. \tag{22}$$

The interaction $S_1$ can be of any form, i.e., be a rather complicated function of fields and their derivatives. In this paper we set a simple goal: refusing from the direct calculation of the integral we consider that the result of the integration can be expressed as a saddle-point equation with a form factor which modulates the action $S_1$. Thus, we assume the following:

$$S_1\left[\bar{\psi}, \psi\right] = S_1\left[n_F\right] = \frac{1}{2} \int d\tau \int d^D r_1 \int d^D r_2\, U\left(r_1 - r_2\right) n_F\left(\tau, r_1\right) n_F\left(\tau, r_2\right). \tag{23}$$

For the condensate density $n_F$:

$$n_F\left(\tau, r\right) = \left[\hat{F}n\right]\left(\tau, r\right) = \int d^D r'\, F\left(r, r'\right) n\left(\tau, r'\right), \tag{24}$$

where $n\left(\tau, r\right) = \bar{\psi}\left(\tau, r\right) \psi\left(\tau, r\right)$. The action of a free theory in this case is:

$$S_0\left[\bar{\psi}, \psi\right] = \int d\tau \int d^D r\, \bar{\psi}\left(\tau, r\right)\left[-i\hbar\partial_\tau - \frac{\hbar^2 \partial_r^2}{2m} + v\left(r\right)\right] \psi\left(\tau, r\right). \tag{25}$$

In order to obtain the mean field equation one has to find the functional derivative $\bar{\psi}$:

$$\frac{\delta n_F\left(\tau, r'\right)}{\delta \bar{\psi}\left(t, r\right)} = \delta\left(\tau - t\right) F\left(r', r\right) \psi\left(t, r\right). \tag{26}$$

After calculations the desired Gross–Pitaevskii equation is obtained (here $r_{12} = r_1 - r_2$):

$$\frac{\delta S\left[\bar{\psi}, \psi\right]}{\delta \bar{\psi}\left(t, r\right)} = \left[-i\hbar\partial_t - \frac{\hbar^2 \partial_r^2}{2m} + v\left(r\right)\right] \psi\left(t, r\right) + \psi\left(t, r\right) \int d^D r_1 \int d^D r_2\, U\left(r_{12}\right) F\left(r_1, r\right) n_F\left(t, r_2\right) = 0. \tag{27}$$

We are interested in stationary solutions (27) thus it is convenient to introduce:

$$\psi\left(t, r\right) = \exp\left(-\frac{i\mu t}{\hbar}\right) \psi\left(r\right), \quad l\left(r\right) = \frac{\hbar^2 \partial_r^2 \psi\left(r\right)}{2m \psi\left(r\right)}. \tag{28}$$

In terms of these denotations the equation in real-space representation has the following form:

$$\int d^D r_1 \int d^D r_2 \int d^D r_3\, U\,(r_{12})\, F\,(r_1, r)\, F\,(r_2, r_3)\, n\,(r_3) = \mu + l\,(r) - v\,(r)\,. \tag{29}$$

Moreover, in momentum-space representation:

$$\int_{k_1} \int_{k_2} U\,(k_1)\, n\,(k_2)\, F\,(k_1, -k_2)\, F\,(-k_1, k) = (2\pi)^D\, \delta^{(D)}\,(k)\, \mu + l\,(k) - v\,(k)\,. \tag{30}$$

Here and further we work in terms of:

$$\int_k = \int \frac{d^D k}{(2\pi)^D}, \quad f\,(r) = \int_k e^{ikr} f\,(k)\,, \quad f\,(k) = \int d^D r\, e^{-ikr} f\,(r)\,. \tag{31}$$

The types of form factors and the solutions of the Gross–Pitaevskii equation with a form factor are discussed below (in the spirit of the books [13,14]), in which a detailed analysis of form factors is carried out within the nonlocal quantum field theory).

### *3.2. Form Factor*

### 3.2.1. The Zoology of Form Factors

In order to develop some intuition regarding the role of form factors and formulate the conditions for choosing the latter, let us turn to specific examples. As is known, the simplest type of a function of two variables is a separable realization. A separable symmetric core $F$ is given by:

$$F\,(r_1, r_2) = F\,(r_2, r_1) = f\,(r_1)\, f\,(r_2)\,. \tag{32}$$

Let us introduce the following notations:

$$\bar{U} = \int d^D r_1 \int d^D r_2\, U\,(r_{12})\, f\,(r_1)\, f\,(r_2)\,, \quad \bar{n} = \int d^D r_3\, f\,(r_3)\, n\,(r_3)\,. \tag{33}$$

Let us note that $\bar{n}$ is a linear functional of the density $n$. This fact will be important further in the narration. In terms of notation (33) the Equation (29) takes the form:

$$\mu \psi\,(r) = -\frac{\hbar^2 \partial_r^2 \psi\,(r)}{2m} + [v\,(r) + \bar{U}\bar{n} f\,(r)]\, \psi\,(r)\,. \tag{34}$$

Thus, our first conclusion is that for a separable form factor the Equation (29) takes the form of an ordinary (linear) Schrödinger equation with an effective potential equal to the sum of the initial and form factor additions, the latter itself contains the integral of the desired solution. If a solution of the Equation (34) is found, one has to use the second equality in (33) and, having obtained the matching condition, find the value $\bar{n}$. Here we note that the role of the trap $v$ can be reduced in favor of the form factor $f$. The next example is the translation-invariant symmetric core $F$ which is given by:

$$F\,(r_1, r_2) = F\,(r_2, r_1) = F\,(r_1 - r_2)\,. \tag{35}$$

In this case, it is convenient to work in the $k$-representation (the momentum representation is the most suitable for translation-invariant problems), since in this representation all the expressions will be simple and clear. The expression for this form factor has the form of

$$F\,(k_1, k_2) = (2\pi)^D\, \delta^{(D)}\,(k_1 + k_1)\, F\,(k_1)\,. \tag{36}$$

If $l$ is considered to be given the Equation (30) becomes algebraic:

$$U(k)\,n(k)\,|F(k)|^2 = (2\pi)^D\,\delta^{(D)}(k)\,\mu + l(k) - v(k). \tag{37}$$

This equation is simple to solve and the expression for $n$ is determined by the reverse transformation to the $r$-representation. If $l$ is given (in Thomas–Fermi approximation $l$ is equal to zero) the expression for the density is obtained.

For further purposes, this material is the hint to answer the question of the final choice of the form factor. Clearly all of that is not a strict proof, but rather a hint to further action. With this knowledge, more complex classes of form factors can be considered, which is done further.

### 3.2.2. Example of Translational Invariance Violation

Outside the translation-invariant case, many beautiful ways of violation of this invariance can be proposed. We begin by considering the core of the form factor in the $k$-representation of the type of (36), but now the Dirac delta function will choose a non-zero value for the total momentum $k_1 + k_2$:

$$F(k_1, k_2) = \sum_{a=1}^{N} f_a(k_1, k_2)\,(2\pi)^D\,\delta^{(D)}(k_1 + k_2 - p_a). \tag{38}$$

In this case the left-hand side of the Equation (30) says:

$$\text{lhs}(k) = \sum_{a,b=1}^{N} U(k - p_a)\,n(k - p_a - p_b)\,f_a(p_a - k, k)\,f_b(k - p_a, p_a + p_b - k). \tag{39}$$

This expression is overloaded for the primary analysis, and the simplest configuration of the parameters has to chosen in (38). Let us choose to momenta ($N = 2$) in a symmetric way: $p_1 = p$ and $p_2 = -p$, and consider the amplitude function $f$ to be constant, $f_a(k_1, k_2) = f_0$. The left-hand side of the Gross–Pitaevskii equation in the $k$-representation will take a form of:

$$\text{lhs}(k) = U(k - p)\,f_0^2\,[n(k - 2p) + n(k)] + U(k + p)\,f_0^2\,[n(k + 2p) + n(k)]. \tag{40}$$

The expression (40) is a complex functional relation, since it contains the density $n$ taken at three different points. Fortunately, such complexity is illusory. To see this we return to the $r$-representation with $2f_0 = 1$. The form factor $F$ takes the form of:

$$F(r_1, r_2) = \cos(pr_1)\,\delta^{(D)}(r_1 - r_2). \tag{41}$$

The Gross–Pitaevskii Equation (29) now contains an integration with respect to only one radius vector $r'$:

$$\cos(pr)\int d^D r'\,U(r - r')\cos(pr')\,n(r') = \mu + l(r) - v(r). \tag{42}$$

Considering the function $l$ to be given, this equation can be solved if we divide all by the cosine appearing on the left-hand side of the equation, and then do the Fourier transform. From the obtained algebraic equation the Fourier image of the product of the desired density and the remaining cosine is determined, and then the inverse transformation is done, which gives:

$$n(r) = \frac{1}{\cos(pr)}\int_k \int d^D r'\,e^{ik(r-r')}\,\frac{\mu - v(r')}{U(k)\cos(pr')}. \tag{43}$$

An important conceptual moment of our paper follows from the Equation (43): up to some details, the form factor of type (41) is most preferable, since it contains all the necessary features for modeling of roton physics. As it will be shown further, the expression (43) remains valid on a qualitative level

for a more general class of quasilocal form factors. Here we note that the *k*-dependent form factors do not generate similar answers.

To conclude the subsection, we note the following: in order to avoid the zero in the denominator (inserted there by us), we can select (using momenta $p_a$) smoother behavior of the integrand, for example:

$$n\left(r\right) = \frac{1}{\Delta^2 + \left[\cos\left(pr\right)\right]^2} \int_k \int d^D r'\, e^{ik(r-r')} \frac{\mu - v\left(r'\right)}{U\left(k\right)\left\{\Delta^2 + \left[\cos\left(pr'\right)\right]^2\right\}}. \tag{44}$$

This is the behavior that should be targeted because it follows from the most general considerations. However, before proving the last statement we derive the Gross–Pitaevskii equation in the class of quasilocal form factors and solve it.

### 3.3. General Solution of the Gp Equation with a Quasilocal Form Factor

Below a general solution of the Equation (29) for a quasilocal form factor (symmetric in both arguments) will be obtained. The latter is an operator function $F$ from $r$ and $\partial_r$ acting on the Dirac delta function:

$$F\left(r_1, r_2\right) = F\left(r_1, \partial_{r_1}\right) \delta^{(D)}\left(r_1 - r_2\right) = F\left(r_2, \partial_{r_2}\right) \delta^{(D)}\left(r_1 - r_2\right). \tag{45}$$

For this particular form factor the stationary Gross–Pitaevskii equation in *r*-representation contains an integration with respect to only one radius vector $r'$:

$$F\left(r, \partial_r\right) \int d^D r'\, U\left(r - r'\right) n_F\left(r'\right) = \mu + l\left(r\right) - v\left(r\right). \tag{46}$$

The Equation (46) can be rewritten in the form:

$$\int d^D r'\, U\left(r - r'\right) n_F\left(r'\right) = F^{-1}\left(r, \partial_r\right) \text{rhs}\left(r\right) \equiv \text{rhs}_F\left(r\right). \tag{47}$$

The obtained equation can be solved with respect to the modulated density $n_F$ by doing the Fourier transform. In the *k*-representation the Equation (47) transforms into the algebraic analog the solution of which is found automatically:

$$n_F\left(k\right) = \frac{\text{rhs}_F\left(k\right)}{U\left(k\right)}. \tag{48}$$

Now one should go back to the *r*-representation and then express the density $n$ in terms of $n_F$. As a result $n$ is given by:

$$n\left(r\right) = F^{-1}\left(r, \partial_r\right) n_F\left(r\right) = \int_k e^{ikr} \frac{\text{rhs}_F\left(k\right)}{U\left(k\right) F\left(r, ik\right)}. \tag{49}$$

Finally, the final equation for the condensate density function $n$ is given by:

$$n\left(r\right) = \int_k \int d^D r'\, e^{ik(r-r')} \frac{\mu - v\left(r'\right)}{U\left(k\right) F\left(r, ik\right) F\left(r', -ik\right)}. \tag{50}$$

The expression (50) has a remarkably simple analytical form, from which we can see all particular cases obtained in the process of obtaining the general solution. Here we should also mention the important property of the solution (50): if the operator function $f$ contains a dependence on $r$, as in the case of a separable form factor, the trap $v$ does not play a primary role.

The obtained general solution (50) still contains a large arbitrariness in the form of a function $F$. To eliminate this arbitrariness, we will now prove some general statements about form factors, and, having determined the final form of the form factor, calculate the integral for the density $n$.

### 3.4. The Choice of the Form Factor

Let us return to the very beginning of our path – the expression for the interaction action $S_1$ and formulate a more general case than the one considered above. Let the form factor modulate not the density $n$, but the fields themselves $\bar{\psi}$. In this case, the expression for the $S_1$ action is given by:

$$S_1 [\bar{\psi}, \psi] = S_1 [N_F] = \frac{1}{2} \int d\tau \int d^D r_1 \int d^D r_2 \, U(r_1 - r_2) \, N_F(\tau, r_1) \, N_F(\tau, r_2). \tag{51}$$

At the same time $N_F(\tau, r)$ is given by:

$$N_F(\tau, r) = \bar{\psi}_F(\tau, r) \, \psi_F(\tau, r) = [\hat{\bar{F}}\bar{\psi}](\tau, r) \, [\hat{F}\psi](\tau, r). \tag{52}$$

Since a description in terms of a functional integral allows for (functional) change of variables under the integral sign, the definition of primary fields is not unique. We turn to the description of our theory in terms of modulated fields (this corresponds to the simplest change of variables). In other words, now let $\psi_F$ and $\bar{\psi}_F$ be new independent fields (denote them as $\varphi$ and $\bar{\varphi}$). In this case, the Gaussian action is rewritten as follows:

$$S_0 [\bar{\varphi}, \varphi] = \int d\tau \int d^D r \, \bar{\varphi}(\tau, r) \, [\hat{G}_\tau^{-1} \varphi](\tau, r). \tag{53}$$

The expression (53) contains a new Gaussian propagator defined by the following expression:

$$\hat{G}_\tau^{-1} = \hat{\bar{F}}^{-1} \left[ -i\hbar \partial_\tau + \hat{H} \right] \hat{F}^{-1}. \tag{54}$$

The last expression is extremely important conceptually: it shows that for the reverse form factor the assumption of analyticity is natural (otherwise we get a propagator that does not have the limit of free theory).

Now we use the obtained experience in our original problem. Moreover, we will refrain from the $k$-dependence of the form factor entirely. The role of this dependence was demonstrated above, and consisted of running constants in an optical trap. Now we need to use a different degree of freedom, which is the dependence on $r$. In the light of what has been said, let us consider the following class of form factors, which is:

$$F(r, ik) = F(r) = \frac{1}{f(r)}. \tag{55}$$

Based on the experience of the Equation (40), without loss of generality we can assume that the function $f$ depends on one external momentum $p$ in terms of the scalar product $pr$. This dependence can be organized in several ways but it is technically easier to assume that this dependence is provided by the exponent. It is also convenient to single out the dependence on two exponents which differ in the sign of argument. Finally, the function $f$ must be expanded into a (double) series in terms of its arguments (they should be considered independent when expanding):

$$f(r) = f\left(e^{ipr}, e^{-ipr}\right) = \sum_{n,m=0}^{\infty} f_{n,m} e^{i(n-m)pr}, \quad f_{n,m} = f_{m,n}. \tag{56}$$

Using the expansion (56) the integral in the Equation (50) for $n$ is calculated in general form in terms of a double converging series. The Equation (50) may be rewritten in the form:

$$n(r) = f(r) \int_k \int d^D r' \, e^{ik(r-r')} \frac{f(r') [\mu - v(r')]}{U(k)}. \tag{57}$$

Then substituting (56) into (57), we get a simple integration of the infinite sum of the Dirac delta functions, after which we get the desired answer for $n$. A remarkable property of the latter is that, even for the zero trap, we obtain a natural model of the roton condensate, which is given by:

$$n\left(\boldsymbol{r}\right) = \mu f\left(\boldsymbol{r}\right) \int_{\boldsymbol{k}} \int d^D \boldsymbol{r}' \, e^{i\boldsymbol{k}(\boldsymbol{r}-\boldsymbol{r}')} \frac{f\left(\boldsymbol{r}'\right)}{U\left(\boldsymbol{k}\right)} = \mu f\left(\boldsymbol{r}\right) \sum_{n,m=0}^{\infty} \frac{f_{n,m} e^{i(n-m)\boldsymbol{pr}}}{U\left[(n-m)\,\boldsymbol{p}\right]}. \tag{58}$$

The expression (58) converges in the case of a regular denominator, since one can construct a majorant for this denominator. Thus, the second equality in (58) reflects a mathematically correct answer for the condensate density. This is one of the main results of our paper.

Let us derive a simplified analogue of the expression (58). The double series appearing in (58) can be simplified if we assume that the function $f$ depends, for example, on the sum of the arguments (hence, on the cosine of the scalar product $\boldsymbol{pr}$):

$$f\left(\boldsymbol{r}\right) = f\left[2\cos\left(\boldsymbol{pr}\right)\right] = \sum_{n=0}^{\infty} f_n \left[2\cos\left(\boldsymbol{pr}\right)\right]^n = \sum_{n=0}^{\infty} \sum_{m=0}^{n} f_n C_n^m e^{i(n-2m)\boldsymbol{pr}}. \tag{59}$$

The difficulty level of the derivation of the expression (59) is the knowledge of binomial expansion. For the zero trap now the following analog of expression (58) is given by:

$$n\left(\boldsymbol{r}\right) = \mu f\left(\boldsymbol{r}\right) \sum_{n=0}^{\infty} \sum_{m=0}^{n} \frac{f_n C_n^m e^{i(n-2m)\boldsymbol{pr}}}{U\left[(n-2m)\,\boldsymbol{p}\right]}, \quad C_n^m = \frac{n!}{m!\,(n-m)!}. \tag{60}$$

Like the expression (58), the equality (60) should be considered one of the main results of this paper. As an example, let us consider an exponential form factor. The exponent has good properties for numerical calculations and graphical illustrations. Qualitative conclusions for the density $n$, made using the example of a concrete form factor, will also be valid in the general case. The exponential form factor itself is given by:

$$f\left(\boldsymbol{r}\right) = e^{\xi\left(e^{i\boldsymbol{pr}} + e^{-i\boldsymbol{pr}}\right)} = e^{2\xi\cos(\boldsymbol{pr})}, \quad f_{n,m} = \frac{\xi^{n+m}}{n!m!}, \quad f_n = \frac{\xi^n}{n!}. \tag{61}$$

Now the plots of $n$ for various values of the parameters appearing in (61) can be constructed. However, before doing this, it is necessary to go back to the denominator which is the most important building block of the expressions (58) and (60) and the exponential density realization. To study this denominator, we turn to the next subsection.

Core and Split

We offer in our opinion a more elegant solution, modifying the idea of the core:

$$U(\boldsymbol{k}) \rightarrow U_{\pm}(\boldsymbol{k}) = \frac{1}{2}\left(U(\boldsymbol{k}) + g_1 \pm \sqrt{U(\boldsymbol{k})^2 + g_2^2}\right). \tag{62}$$

Using the idea of a "split" in the expressions for the density of a condensate, we get a series of graphical dependencies, which is given below.

*3.5. Condensate Density*

Here we discuss the behavior of the condensate density as a function of the order of the interaction and the fixed momentum $\boldsymbol{p}$, using the exponential form factor with $\xi = 1$. We also study the the rate of convergence of the series in the expression for the density, using the example of plots. The plots are shown in Figures 3–14. Here $\theta_D$, $\theta_Q$ and $\theta_O$ are the angles at which the corresponding interaction vanishes.

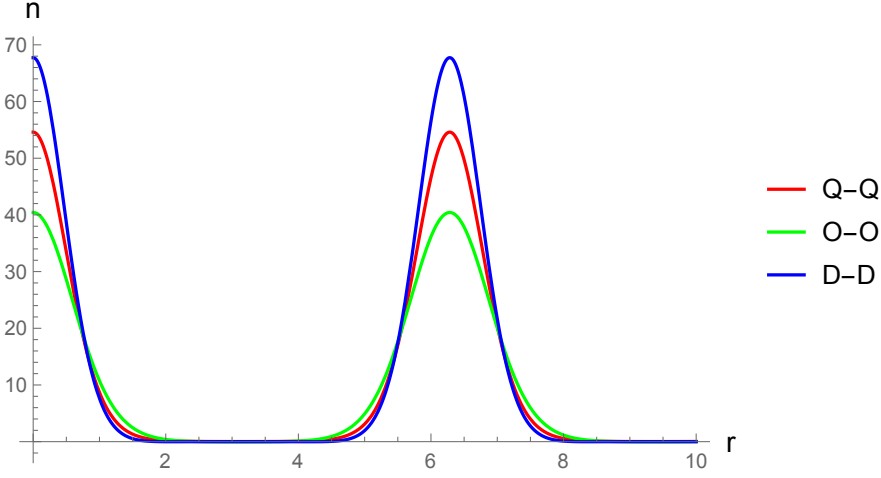

**Figure 3.** $\theta = \theta_D$, $n = 10$, $p = 1$.

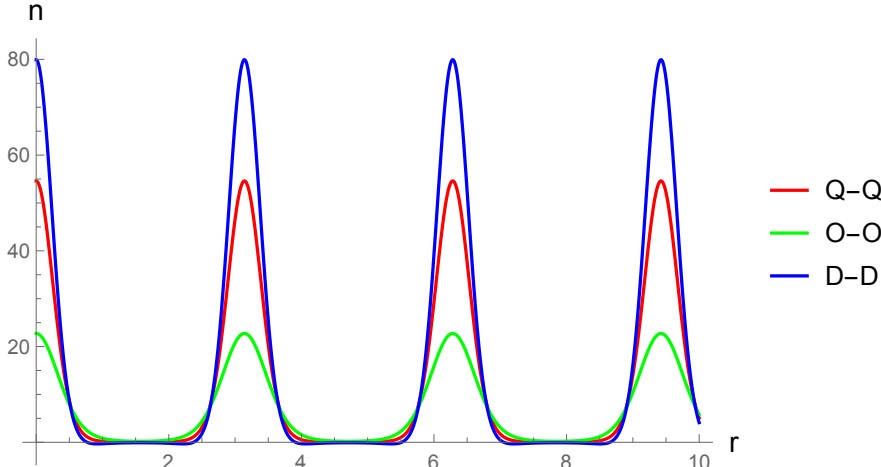

**Figure 4.** $\theta = \theta_D$, $n = 10$, $p = 2$.

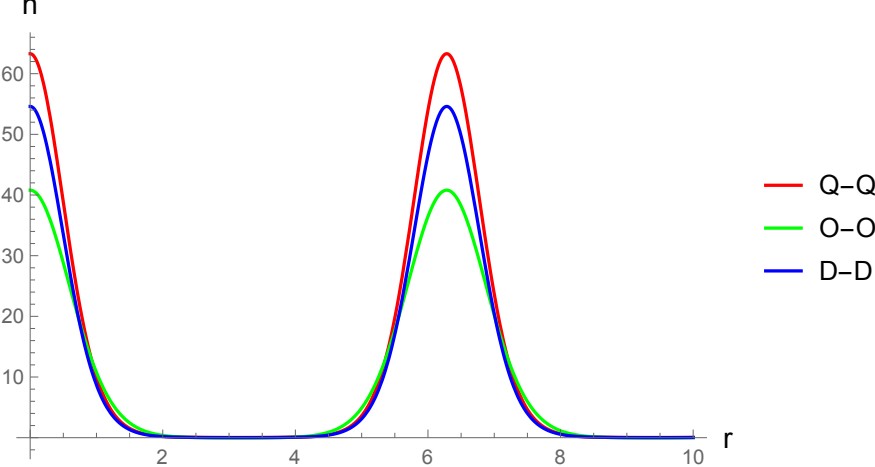

**Figure 5.** $\theta = \theta_Q$, $n = 10$, $p = 1$.

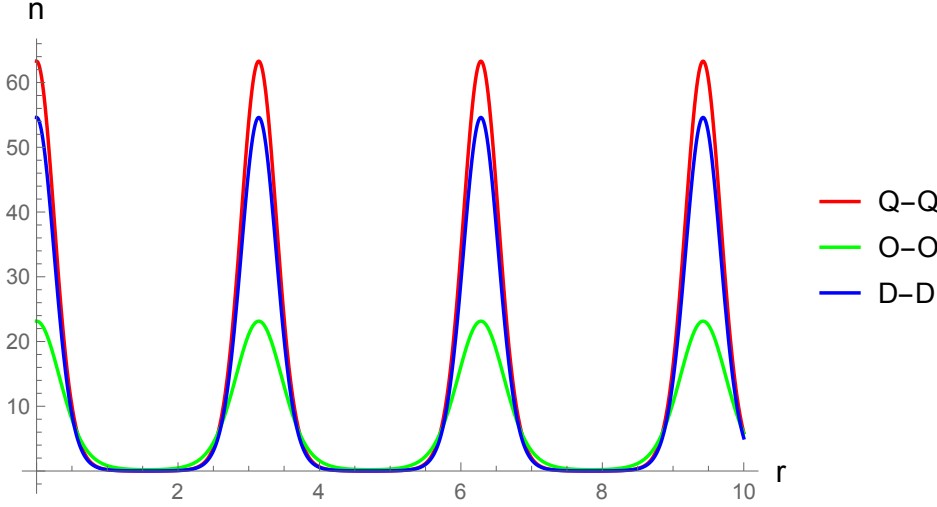

**Figure 6.** $\theta = \theta_Q$, $n = 10$, $p = 2$.

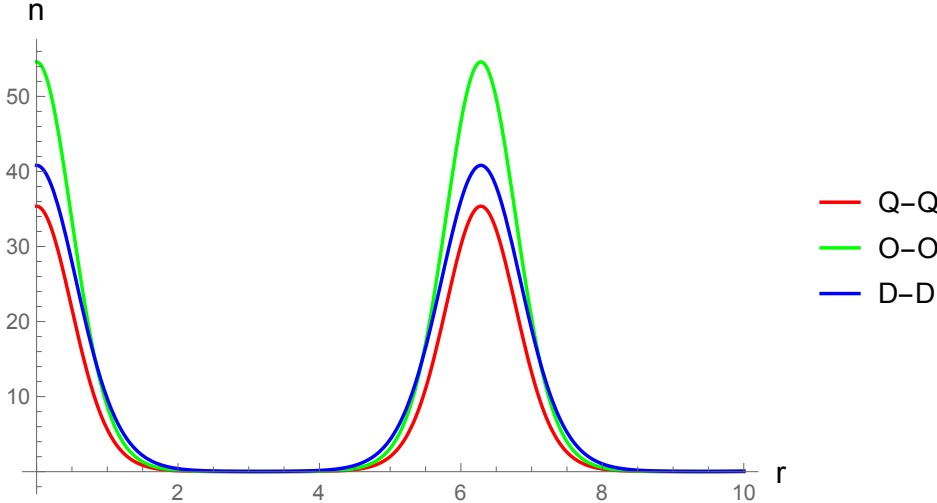

**Figure 7.** $\theta = \theta_O$, $n = 10$, $p = 1$.

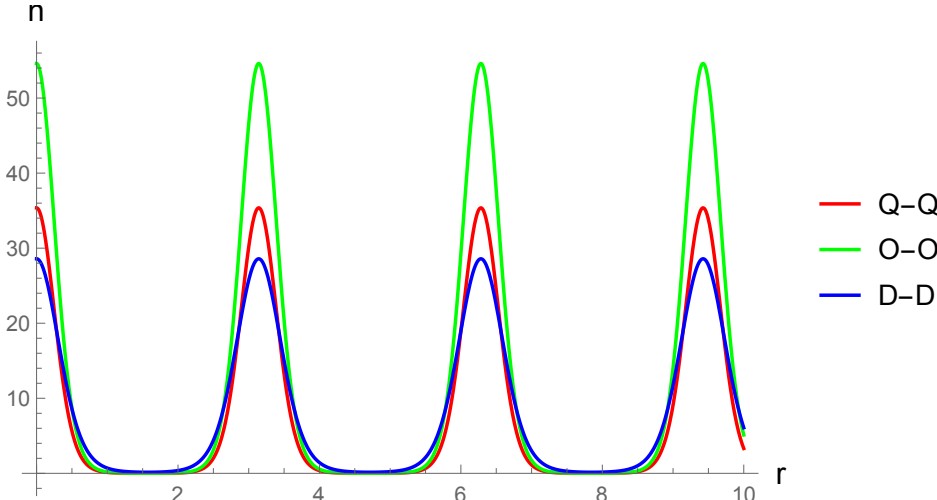

**Figure 8.** $\theta = \theta_O$, $n = 10$, $p = 2$.

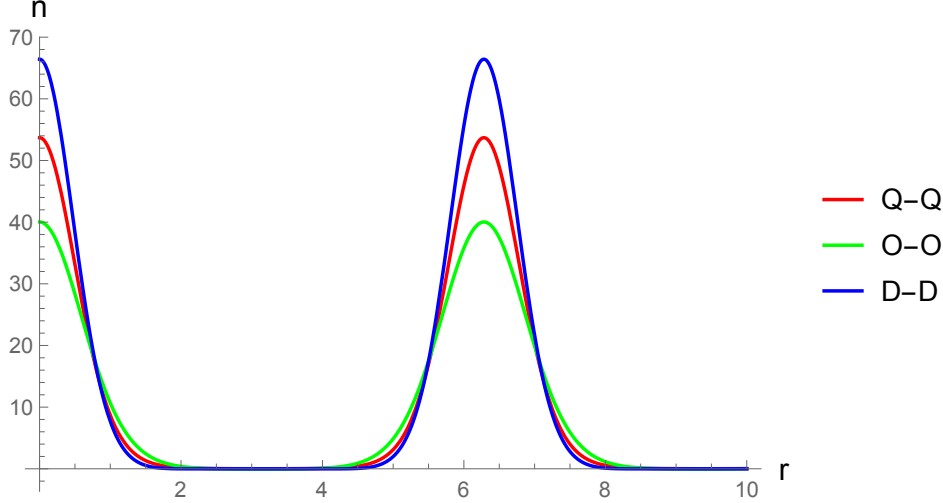

**Figure 9.** $\theta = \theta_D$, $n = 5$, $p = 1$.

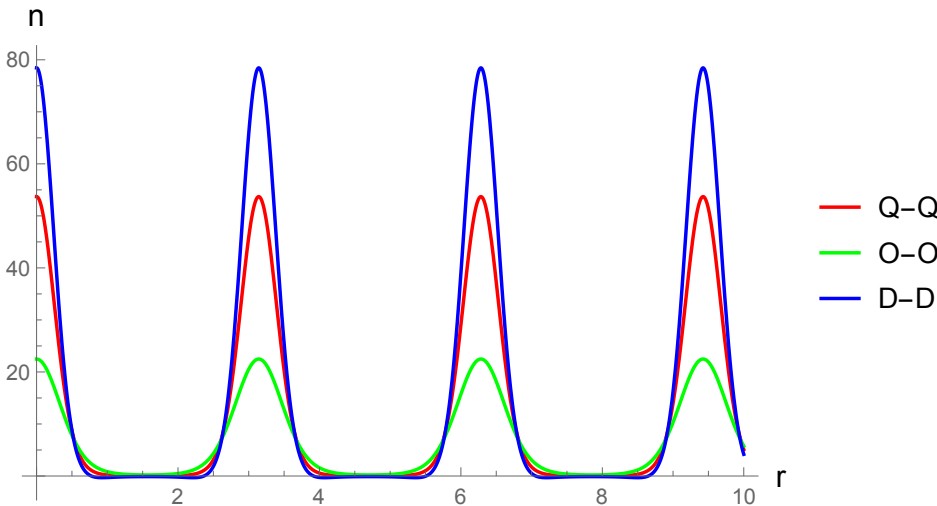

**Figure 10.** $\theta = \theta_D$, $n = 5$, $p = 2$.

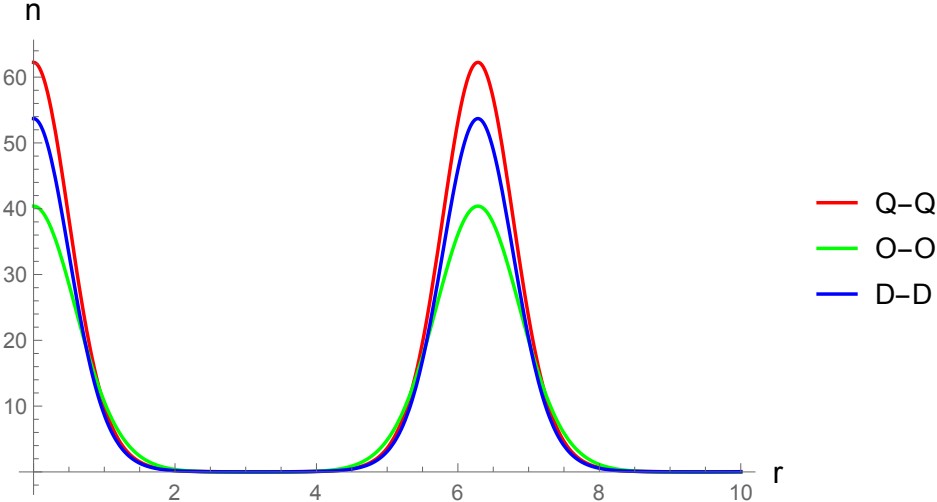

**Figure 11.** $\theta = \theta_Q$, $n = 5$, $p = 1$.

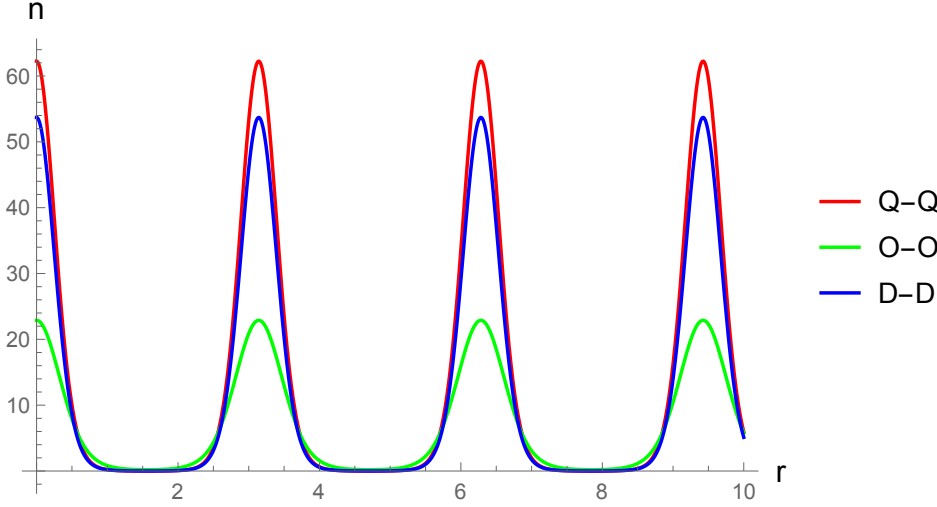

**Figure 12.** $\theta = \theta_Q$, $n = 5$, $p = 2$.

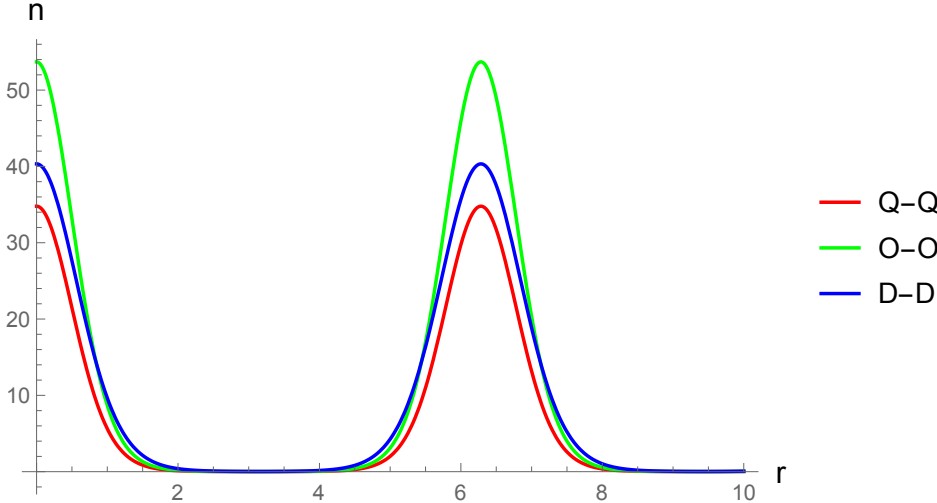

**Figure 13.** $\theta = \theta_O$, $n = 5$, $p = 1$.

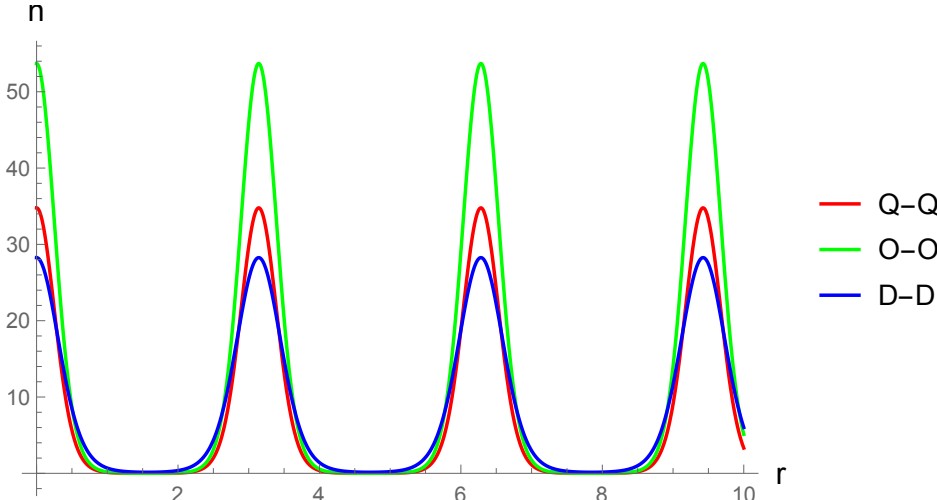

**Figure 14.** $\theta = \theta_O$, $n = 5$, $p = 2$.

This concludes the discussion of the solution of the Gross–Pitaevskii equation with the form factor in the Thomas–Fermi approximation and we turn to the discussion of quadrupoles in the optical lattice.

*3.6. Quadrupoles in the Optical Lattice*

We consider a stationary condensate thus the transition $\psi \to \psi \exp(-i\mu/\hbar)$ is valid, and also in the Gross–Pitaevskii equation due to the smallness of the density the last term can be neglected. The chemical potential $\mu$ can vary continuously, but in the course of the analysis it is shown that the formation of condensate in the low-density limit is possible only in the case of discrete values of $\mu$, similar to the spectrum of bound states when considering the single-particle Schrödinger equation.

3.6.1. Where Do Small Concentrations Lead to?

In case of small concentrations the Gross–Pitaevskii equation takes the form:

$$\mu\psi(\boldsymbol{r}) = -\frac{\hbar^2}{2m}\Delta\psi(\boldsymbol{r}) + v(\boldsymbol{r})\psi(\boldsymbol{r}). \tag{63}$$

Consider a trap with an optical potential of a special form:

$$v(\boldsymbol{r}) = -A_1 \sin^2(\boldsymbol{k}_1\boldsymbol{r}) - A_2 \sin^2(\boldsymbol{k}_2\boldsymbol{r}) - A_3 \sin^2(\boldsymbol{k}_3\boldsymbol{r}), \tag{64}$$

where $\boldsymbol{k}_1$, $\boldsymbol{k}_2$, $\boldsymbol{k}_3$ are the vectors along the $x$, $y$, $z$ axis respectively. The justification for the chosen trap is as follows: we consider such a trap, since in the limit of a small $\boldsymbol{k}_i\boldsymbol{r} \to 0$ the given potential transforms into the ordinary (anisotropic) harmonic potential, which is:

$$v(\boldsymbol{r}) \approx -\sum_i^3 A_i(\boldsymbol{k}_1\boldsymbol{r})^2. \tag{65}$$

Besides the potential can be represented as a sum of a form:

$$v(\boldsymbol{r}) = v_1(x) + v_2(y) + v_3(z), \tag{66}$$

where each term depends only on one coordinate. Due to this, we can search for the wave function in the form of $\psi(\boldsymbol{r}) = \psi_1(x)\psi_2(y)\psi_3(z)$, in other words, use the method of separation of variables. Then, writing out the Laplace operator and using $\mu = \mu_1 + \mu_2 + \mu_3$, three equations are obtained:

$$\left(-\frac{\hbar^2}{2m}\partial_i^2 - A_i \sin^2(\boldsymbol{k}_i\boldsymbol{r})\right)\psi_i = \mu_i\psi_i. \tag{67}$$

These equations can be reduced to the following form:

$$\left(-\frac{\hbar^2}{2m}\partial_i^2 + \frac{A_i}{2}\cos(2\boldsymbol{k}_i\boldsymbol{r})\right)\psi = \left(\mu_i + \frac{A_i}{2}\right)\psi_i. \tag{68}$$

Thus, the Gross–Pitaevskii equation splits into three one-dimensional Schrödinger equations in a periodic potential.

3.6.2. Schrödinger Equation in a Periodic Potential

We have the Schrödinger equation in a periodic potential. After the transformations, it can be rewritten in the following form (for certainty, consider the $x$-coordinate and omit the indices):

$$\psi'' + \frac{m}{\hbar^2}\left((A + 2\mu) - A\cos(2kx)\right)\psi = 0. \tag{69}$$

We do the change of variables $z = kx$ and get:

$$\ddot{\psi}(z) + \frac{m}{\hbar^2 k^2}\left((A + 2\mu) - A\cos(2z)\right)\psi(z) = 0, \tag{70}$$

where the dot denotes a $z$ derivative. Then we introduce notations $E(k, \mu) \equiv m(A + 2\mu)/\hbar^2 k^2$, $h^2(k) \equiv mA/2\hbar^2 k^2$. Here, the parametric dependence of $E$ and $h$ on the preselected momentum $k$ is emphasized. With these notations taken into account, the equation takes a very compact form:

$$\ddot{\psi} + (E - 2h^2\cos 2z) = 0. \tag{71}$$

However, this form is deceptive: The obtained equation is the Mathieu equation, which is very difficult for analytical analysis.

Now let us consider the change $z \to z + \pi$. Since the cosine value will not change, we have two proportional solutions $\psi(z)$ and $\psi(z + \pi)$. This can be written in the form:

$$\hat{T}_\pi\psi(z) = f\psi(z + \pi), \quad f = \text{const.} \tag{72}$$

If we consider $z = 0$ and $z = \pi$ an important expression $f^2 = \psi(2\pi)/\psi(0) = 1$ is found. Now the case of small values of the parameter $h$ can be considered.

### 3.6.3. Weak Coupling Mode

If the parameter $h$ is small the Mathieu equation is given by:

$$\ddot{\psi} + E\psi = 0. \tag{73}$$

It is obvious that the general solution of this equation is the function $\psi = C_1\sin(\sqrt{E}z) + C_2\cos(\sqrt{E}z)$. Let us denote $\psi_+ = a\cos(\sqrt{E}z)$ and $\psi_- = b\sin(\sqrt{E}z)$. In this case the operator $\hat{T}_\pi$ acts as follows:

$$\psi(z + \pi) = f(\alpha\psi_+ + \beta\psi_-) = f[\alpha a]. \tag{74}$$

Using the solutions for $\psi_+$ and $\psi_-$, and also the action of the translation operator on them the Wronskian of these solutions can be constructed, and $W[\psi_+, \psi_-] = ab\sqrt{E}$. Having written the Wronskian in the explicit form, a quadratic equation for $f$ is obtained. Its solutions are given by:

$$f_\pm = \frac{2b\sqrt{E}\psi_+(\pi) \pm \sqrt{2b\sqrt{E}\psi_+(\pi) - 4a^2b^2E}}{2ab\sqrt{E}}. \tag{75}$$

Besides, $f_+ = 1/f_-$. Saying that $f_+ = \exp(i\pi\nu)$, we get that:

$$f_+ + f_- = 2\cos\pi\nu = \frac{2\psi_+(\pi)}{a}, \tag{76}$$

i.e., $\nu = \sqrt{E}$. Then we find that:

$$\exp(2i\sqrt{E}\pi) = 1. \tag{77}$$

This shows that $E = s^2$, where $s$ is an integer. As a result a restriction on $\mu$ is obtained:

$$\mu = \frac{\hbar^2 k^2 s^2}{2m} - \frac{A}{2}. \tag{78}$$

Thus, in the weak-coupling mode (for small values of $h$) the condensate wave functions match with the wave functions of the particle in the potential well. The last unknown constant is calculated from the normalization of the wave function:

$$\int |\psi^2|\, dV = N. \tag{79}$$

We investigated the case of small values of $h$ and now turn to the case of strong coupling mode, i.e., large values of $h$.

### 3.6.4. Strong Coupling Mode

The potential $2h^2 \cos(2z)$ has its minimum in $z = \pm\pi/2$ and we will use its expansion. Thus it is convenient to rewrite the Mathieu equation as follows:

$$\psi'' + \left(E + 2h^2 \cos\left(2z \pm \pi\right)\right)\psi = 0. \tag{80}$$

Using the expansion

$$\cos\left(2z \pm \pi\right) \approx 1 - 2\left(z \pm \frac{\pi}{2}\right)^2, \tag{81}$$

and introducing a new variable

$$\xi = 2\sqrt{h}\left(z \pm \frac{\pi}{2}\right), \tag{82}$$

the Mathieu equation takes the form of

$$\frac{d^2\psi}{d\xi^2} + \left(\frac{E + 2h^2}{4h} - \frac{\xi^2}{4}\right) = 0. \tag{83}$$

This is the Weber equation, the solution of which is a parabolic cylinder function. However, provided that

$$\frac{E + 2h^2}{2h} = 2s + 1, \tag{84}$$

parabolic cylinder functions $D_s$ become $\exp(-x^2/4)\mathrm{He}_s$, where $\mathrm{He}_s$ is a modified Hermite polynomial. Therefore the energy spectrum has the form of a harmonic oscillator spectrum and $\mu$ must satisfy the following condition:

$$\mu = \frac{\hbar k}{2}\sqrt{\frac{A}{m}}\left(s + \frac{1}{2}\right) - A. \tag{85}$$

The constant is calculated from the normalization of the wave function.

### 3.6.5. Comparative Analysis of Weak and Strong Coupling Modes

In the course of the study of the potential $v(r) = -\sum_i A_i \sin^2(k_i r)$ we obtained different spectra of the chemical potential in the case of small and large values of parameters $A_i$:

$$\mu_w = \frac{\hbar^2 k^2 s^2}{2m} - \frac{A}{2}; \quad \mu_s = \frac{\hbar k}{2}\sqrt{\frac{A}{m}}\left(s + \frac{1}{2}\right) - A, \tag{86}$$

where the index $w$ denotes the weak coupling mode, and the index $s$ denotes the strong coupling mode. Schematically, the difference between the strong and weak coupling can be represented by the Figure 15.

This figure shows the different modes of quantizing of the chemical potential from linear to quadratic. It can also be shown that the spectrum of the chemical potential has a fine structure, and this splitting of the levels makes it possible to connect the strong and weak coupling modes. In this case, the splitting of the levels is proportional to $\exp(-h)$. This study is given in the work [34], and it is

classical, for this reason it is also given in the monograph [35] devoted to the course of quantum mechanics. This completes the section "The Gross–Pitaevskii Equation" and we proceed to the analysis of the condensate excitations.

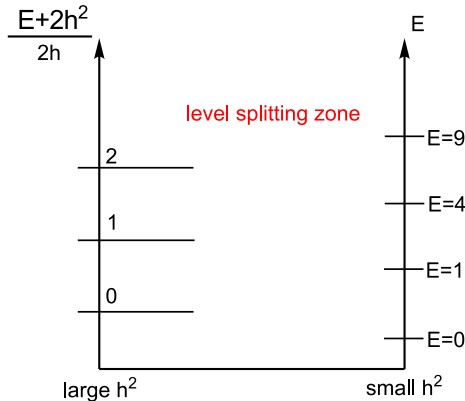

**Figure 15.** The structure of the spectra in the case of small and large values of $h^2$.

## 4. Analysis of Condensate Excitations

In this section we consider condensate excitations in our model and look for their spectrum using the Gross–Pitaevskii functional. Further in the section, we study the dependence of the critical momentum, destroying the stability of solutions of the Gross–Pitaevskii equation, on the condensate momentum. The analysis begins with a revision of the main aspects of the already described material.

### 4.1. Intermediate Results

The action of the free theory $S_0$ is given by:

$$S_0 [\bar{\psi}, \psi] = \int d\tau \int d^D r \, \bar{\psi} (\tau, r) \left[ -i\hbar \partial_\tau + \hat{H} \right] \psi (\tau, r). \tag{87}$$

The Hamiltonian has a standard form of the sum of kinetic energy and the potential of the trap:

$$\hat{H} = H (r, \partial_r) = \varepsilon (\partial_r) + v (r), \quad \varepsilon (\partial_r) = -\frac{\hbar^2 \partial_r^2}{2m}, \quad \partial_r = \frac{\partial}{\partial r}. \tag{88}$$

The equation for the interaction action $S_1$ is given by:

$$S_1 [\bar{\psi}, \psi] = \frac{1}{2} \int d\tau \int d^D r_1 \int d^D r_2 \, \Gamma (r_1, r_2) \, n (\tau, r_1) \, n (\tau, r_2). \tag{89}$$

In the last equation the effective (modulated by the form factor) interaction $\Gamma$ and the density $n$ are introduced:

$$\Gamma (r_1, r_2) = \Gamma (r_2, r_1) = \frac{U (r_1 - r_2)}{f (r_1) f (r_2)}, \quad n (\tau, r) = \bar{\psi} (\tau, r) \psi (\tau, r). \tag{90}$$

For the field $\psi$ we choose an ansatz in which it is the sum of a stationary solution of the Gross–Pitaevskii equation in the Thomas–Fermi approximation and a small perturbation $\delta \psi$:

$$\psi (t, r) = \psi_0 (t, r) + \delta \psi (t, r), \quad \psi_0 (t, r) = e^{-\frac{i\mu t}{\hbar}} \psi_0 (r). \tag{91}$$

At the same time we look for the perturbation $\delta \psi$ in the following form:

$$\delta \psi (t, r) = e^{-\frac{i\mu t}{\hbar}} \rho (t, r), \quad \rho (t, r) = e^{-i\omega t} A (r) + e^{i\omega t} B (r). \tag{92}$$

We also need the solution of the Gross–Pitaevskii equation itself which in this section is given by:

$$[i\hbar\partial_t - H(\mathbf{r}, \partial_{\mathbf{r}})]\,\psi(t,\mathbf{r}) = \psi(t,\mathbf{r})\int d^D\mathbf{r}'\,\Gamma(\mathbf{r},\mathbf{r}')\,n(t,\mathbf{r}')\,. \tag{93}$$

In the next subsection, the Gross–Pitaevskii functional for a specific form factor will be obtained, which will later be used for analysis.

### 4.2. Derivation of the Gp Functional for a Form Factor of a Special Form

Substituting the above ansatz for $\psi$ and $\delta\psi$ into expressions for the action of the free theory $S_0$, the difference $\Delta S_0$ of the action on the perturbed and stationary field configurations:

$$\begin{aligned}
\Delta S_0[\bar{\psi}, \psi] &= \int d\tau \int d^D\mathbf{r}\,\delta\bar{\psi}(\tau,\mathbf{r})\left[-i\hbar\partial_\tau + \hat{H}\right]\delta\psi(\tau,\mathbf{r}) + \\
&\quad \int d\tau \int d^D\mathbf{r}\,\delta\bar{\psi}(\tau,\mathbf{r})\left\{\left[-i\hbar\partial_\tau + \hat{H}\right] + \left[-i\hbar\partial_\tau + \hat{H}\right]^T\right\}\psi_0(\tau,\mathbf{r})\,.
\end{aligned} \tag{94}$$

This seemingly cumbersome expression is greatly simplified when taking into account periodic boundary conditions, which give:

$$\int_{t_1}^{t_2} d\tau\, e^{\pm i\omega\tau} = \mp\frac{i}{\omega}\left(e^{\pm i\omega t_2} - e^{\pm i\omega t_1}\right) = 0. \tag{95}$$

Such boundary conditions lead to the fact that all linear terms in $\delta\psi$ do not contribute to the difference of actions, from which it follows that they do not contribute to the Gross–Pitaevskii functional also. Substituting the same ansatz into the expression for the interaction action $S_1$ and again calculating the difference on the perturbed and stationary field configurations in the Gaussian approximation, we get the expression for the difference $\Delta S_1$:

$$\begin{aligned}
\Delta S_1[\bar{\psi}, \psi] &\equiv S_1[\bar{\psi}, \psi] - S_1[\bar{\psi}_0, \psi_0] = \\
&\frac{1}{2}\int d\tau \int d^D\mathbf{r}_1 \int d^D\mathbf{r}_2\,\Gamma(\mathbf{r}_1,\mathbf{r}_2)\left\{\chi(\tau,\mathbf{r}_1)\chi(\tau,\mathbf{r}_2) + 2n_0(\tau,\mathbf{r}_1)\left[\chi(\tau,\mathbf{r}_2) + \delta\bar{\psi}(\tau,\mathbf{r}_2)\delta\psi(\tau,\mathbf{r}_2)\right]\right\},
\end{aligned} \tag{96}$$

where for brevity

$$\chi(t,\mathbf{r}) = \bar{\psi}_0(t,\mathbf{r})\,\delta\psi(t,\mathbf{r}) + \psi_0(t,\mathbf{r})\,\delta\bar{\psi}(t,\mathbf{r})\,. \tag{97}$$

The difference of the actions $\Delta S_1$ is simplified due to periodic boundary conditions (linear in $\delta\psi$ do not contribute to $\Delta S_1$), and also due to the Gross–Pitaevskii equation, which gives:

$$\int d\tau \int d^D\mathbf{r}_1 \int d^D\mathbf{r}_2\,\Gamma(\mathbf{r}_1,\mathbf{r}_2)\,n_0(\tau,\mathbf{r}_1)\,\delta\bar{\psi}(\tau,\mathbf{r}_2)\,\delta\psi(\tau,\mathbf{r}_2) = \int d\tau \int d^D\mathbf{r}\,\delta\bar{\psi}(\tau,\mathbf{r})\left[\mu - v(\mathbf{r})\right]\delta\psi(\tau,\mathbf{r})\,. \tag{98}$$

When substituting the phase exponents, this term cancels out with the same in the difference of actions $\Delta S_0$. Thus the Gross–Pitaevskii functional is given by:

$$\begin{aligned}
\Delta S[\bar{\psi}, \psi] &= \int d\tau \int d^D\mathbf{r}\,\delta\bar{\psi}(\tau,\mathbf{r})\left[\mu - i\hbar\partial_\tau + \varepsilon(\partial_{\mathbf{r}})\right]\delta\psi(\tau,\mathbf{r}) \\
&+ \frac{1}{2}\int d\tau \int d^D\mathbf{r}_1 \int d^D\mathbf{r}_2\,\Gamma(\mathbf{r}_1,\mathbf{r}_2)\,\chi(\tau,\mathbf{r}_1)\chi(\tau,\mathbf{r}_2)\,.
\end{aligned} \tag{99}$$

Implementing the substitutions left for $\psi$ and $\delta\psi$ and also rewriting

$$\chi(t,\mathbf{r}) = e^{-i\omega t}C(\mathbf{r}) + e^{i\omega t}\bar{C}(\mathbf{r})\,, \quad C(\mathbf{r}) = \bar{\psi}_0(\mathbf{r})A(\mathbf{r}) + \psi_0(\mathbf{r})\bar{B}(\mathbf{r})\,, \tag{100}$$

by integrating over time we get the desired functional:

$$\Delta S \left[\bar{\psi}, \psi\right] = \int d^D r \left\{\bar{A}\left(r\right)\left[-\hbar\omega + \varepsilon\left(\partial_r\right)\right]A\left(r\right) + \bar{B}\left(r\right)\left[\hbar\omega + \varepsilon\left(\partial_r\right)\right]B\left(r\right)\right\} +$$
$$\int d^D r_1 \int d^D r_2\, \Gamma\left(r_1, r_2\right)\psi_0\left(r_1\right)\psi_0\left(r_2\right)\left[B\left(r_1\right) + \bar{A}\left(r_1\right)\right]\left[A\left(r_2\right) + \bar{B}\left(r_2\right)\right]. \tag{101}$$

Here we also used the fact that the field $\psi_0$ is real. It is the Gross–Pitaevskii functional that we will study further in our paper. With its help the equation for the critical momentum, creating an instability in the solution of the corresponding GP equation, will be derived further.

### 4.3. Derivation of the Equation for the Critical Momentum

The functional derivative of the perturbation is given by:

$$A\left(r\right) = \frac{u_k}{\sqrt{V}}e^{ikr}, \quad B\left(r\right) = \frac{\bar{v}_k}{\sqrt{V}}e^{-ikr}, \quad \varepsilon\left(\partial_r\right) \to \varepsilon_k = \frac{\hbar^2 k^2}{2m}. \tag{102}$$

In this case the Gross–Pitaevskii functional is given by

$$\Delta S\left[\bar{\psi}, \psi\right] = \left[-\hbar\omega + \varepsilon_k\right]\bar{u}_k u_k + \left[\hbar\omega + \varepsilon_k\right]\bar{v}_k v_k + I_k\left[\bar{u}_k + \bar{v}_k\right]\left[u_k + v_k\right], \tag{103}$$

where for brevity

$$I_k = \frac{1}{V}\int d^D r_1 \int d^D r_2\, \Gamma\left(r_1, r_2\right)\psi_0\left(r_1\right)\psi_0\left(r_2\right)e^{ik(r_1 - r_2)} =$$
$$\frac{1}{V}\int d^D r_1 \int d^D r_2\, \Gamma\left(r_1, r_2\right)\psi_0\left(r_1\right)\psi_0\left(r_2\right)\cos\left[ik\left(r_1 - r_2\right)\right]. \tag{104}$$

Thus, the problem of analysis of the stability of the Gross–Pitaevskii equation comes down to the problem of finding the Bogoliubov spectrum. The extremum conditions are as follows:

$$\Delta S\left[\bar{\psi}, \psi\right] \equiv S\left(\bar{u}, u, \bar{v}, v\right), \quad \frac{\partial S}{\partial \bar{u}} = \frac{\partial S}{\partial u} = \frac{\partial S}{\partial \bar{v}} = \frac{\partial S}{\partial v} = 0, \tag{105}$$

from which is clear that $\bar{u} = u$ and $\bar{v} = v$, and the equations connecting these to variables are

$$\left[-\hbar\omega + \varepsilon_k + I_k\right]u_k + I_k v_k = 0, \quad I_k u_k + \left[\hbar\omega + \varepsilon_k + I_k\right]v_k = 0. \tag{106}$$

From the condition of the existence of a non-trivial solution for this system it follows that the following equation is valid (the determinant vanishes):

$$\left(\hbar\omega_k\right)^2 = \varepsilon_k\left(\varepsilon_k + 2I_k\right). \tag{107}$$

The obtained equation is the equation for the perturbation spectrum. Now let us derive the equation for the critical momentum, preliminarily transforming the expression for $I$ as follows:

$$I_k = \frac{\mu}{V}\int d^D r_1 \int d^D r_2\, U\left(r_1 - r_2\right)\sqrt{\frac{\Sigma\left(r_1\right)\Sigma\left(r_2\right)}{f\left(r_1\right)f\left(r_2\right)}}\cos\left[ik\left(r_1 - r_2\right)\right]. \tag{108}$$

Here we use the fact that $\psi_0$ is given by:

$$\psi_0\left(r\right) = \sqrt{\mu f\left(r\right)\Sigma\left(r\right)}. \tag{109}$$

The equation for the critical momentum can be written in a compact form:

$$\varepsilon_{k_0} + 2I_{k_0} = 0. \tag{110}$$

However, in the explicit form this equation is rather cumbersome:

$$\frac{1}{V} \int d^D r_1 \int d^D r_2 \, U \left( r_1 - r_2 \right) \sqrt{\frac{\Sigma \left( r_1 \right) \Sigma \left( r_2 \right)}{f \left( r_1 \right) f \left( r_2 \right)}} \cos \left[ i k_0 \left( r_1 - r_2 \right) \right] = -\frac{\varepsilon_{k_0}}{2\mu}. \tag{111}$$

This equation determines, for example, the dependence of the absolute value of $k_0$ on the direction $n_0$. Though we study it in another way. Let us consider the limiting cases of this equation, which allow us to discuss the dependence of $k_0$ on the momentum of the stationary condensate $p$.

### 4.4. Analysis of Condensate Stability

The Equation (111) is quite complicated, so to solve it we use a trick, which is well known from the theory of superconductivity. Let the direction $n_0$ be so that the main contribution is given by attraction (in the case of the attraction the Equation (110) has a solution). In this case, we make the following substitution:

$$U \left( r_1 - r_2 \right) \to g \delta^{(D)} \left( r_1 - r_2 \right). \tag{112}$$

Then the Equation (111) takes a simple for analysis purposes form:

$$\frac{g}{V} \int d^D r \, \frac{\Sigma \left( r \right)}{f \left( r \right)} = \frac{\varepsilon_{k_0}}{2\mu} = \frac{\hbar^2 k_0^2}{4m\mu}. \tag{113}$$

The solution of the obtained equation is given by:

$$k_0^2 \left( p \right) = \frac{4m\mu g}{\hbar^2 V} \int d^D r \, \frac{\Sigma \left( p, r \right)}{f \left( pr \right)}. \tag{114}$$

Then we use the earlier chosen form factor:

$$f \left( r \right) = e^{\xi \left( e^{ipr} + e^{-ipr} \right)} = e^{2\xi \cos(pr)}. \tag{115}$$

We emphasize here that the qualitative conclusions for $k_0$ made using the example of a specific form factor are also valid in the general case. To calculate the integral, we take into account the following equality:

$$\int d^D r \, e^{ipr(n-m+n'-m')} = (2\pi)^D \, \delta^{(D)} \left[ p \left( n - m + n' - m' \right) \right] = V \delta_{n-m+n'-m',0}. \tag{116}$$

To make the transition to the Kronecker delta the following correspondence is used:

$$V = (2\pi)^D \, \delta^{(D)} \left( 0 \right). \tag{117}$$

Now the integral can be calculated:

$$\frac{1}{V} \int d^D r \, \frac{\Sigma \left( p, r \right)}{f \left( pr \right)} = \sum_{n,n',m,m'=0}^{\infty} \frac{\xi^{n+n'+m+m'} \left( -1 \right)^{n'+m'} \delta_{n+n'-m-m',0}}{n! n'! m! m'! U \left[ \left( n - m \right) p \right]} = $$
$$\sum_{n,n'=0}^{\infty} \sum_{m=0}^{n} \sum_{m'=0}^{n'} \frac{\xi^{n+n'} C_n^m C_{n'}^{m'} \left( -1 \right)^{n'} \delta_{n+n'-2m-2m',0}}{n! n'! U \left[ \left( n - 2m \right) p \right]}. \tag{118}$$

Because of the rapid convergence of the series, we can confine ourselves to a few first terms, which is confirmed numerically. After calculating the integral, the dependence of the critical momentum $k_0$ on the condensate momentum $p$ should be considered graphically. Before this, it is important to note that in the case of $D = 3$ for D-D interaction $k_0$ does not depend on $p$ because the interaction potential $U_D(k)$ does not depend on the absolute value of the momentum $k$.

The above plots show that the dependence has an asymptote, which was expected, because the Q-Q interaction $\propto k^2$ and the O-O interaction $\propto k^4$. In the case of quadrupoles for small $p$, we obtain that $k_0 \gg p$. For octupoles, for a small $p$, the critical momentum is practically constant, but at a certain value of $p$ sharply decreases, reaching a plateau smoothly. Moreover, the indicated dependence is valid for angles $\alpha$ which are not equal to those angles at which the interaction vanishes ($\alpha_Q$ and $\alpha_O$, respectively). Plots of the dependence of the critical momentum $k_0$ on the momentum of the stationary condensate $p$ are shown in Figures 16 and 17.

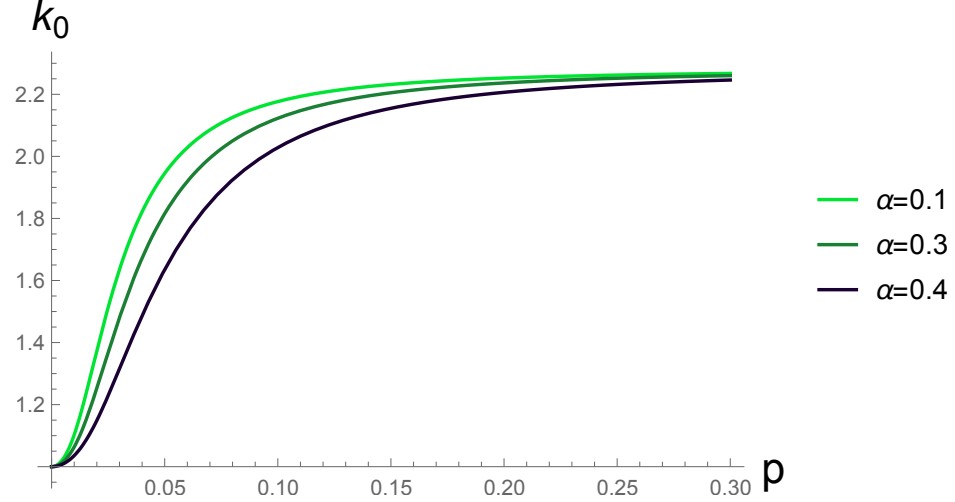

**Figure 16.** The dependence of critical momentum $k_0$ on the momentum of quadrupolar condensate.

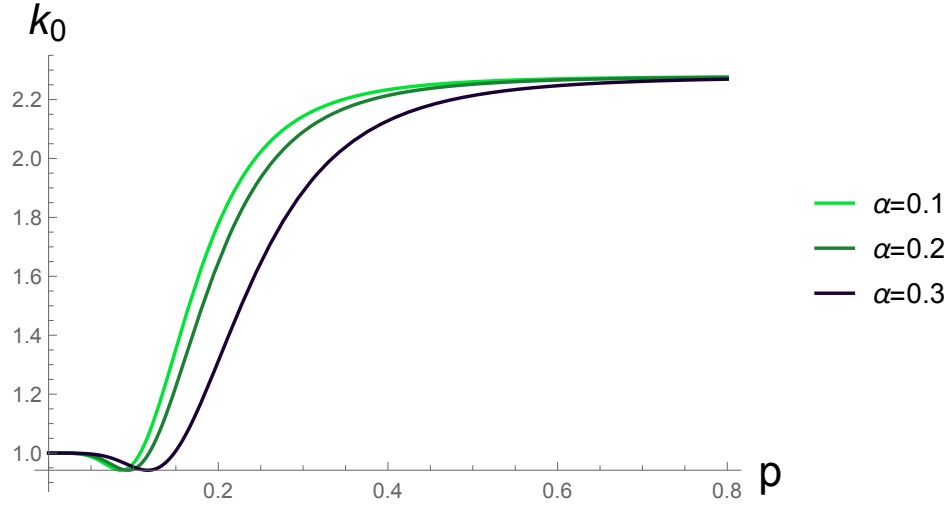

**Figure 17.** The dependence of critical momentum $k_0$ on the momentum of octupolar condensate.

## 5. Conclusions

In this paper the properties of classical and quantum Bose gases of dipoles, axisymmetric quadrupoles and octupoles with different multipole order are studied. Classical interaction potentials in corresponding gases in the coordinate and momentum representations were calculated. In this case, various modifications of this interaction are discussed in details, which are the classical modification using core and the original

modification using "split" – splitting the interaction by a certain feature. In this paper it is the sign of the corresponding interaction range (this representation is very convenient if we want, for example, to investigate the threshold of instability in the considered gas).

Next, the quantum theory of Bose gases of dipoles, axisymmetric quadrupoles and octupoles in the Gross–Pitaevskii formalism is discussed. The starting point of this consideration is the Gross–Pitaevskii functional, from which the same name equation is derived in both coordinate and momentum representations. The original point of this paper is that we studied not the "ordinary phonon" condensate and its excitations, but considered a more general scenario for the appearance of a spatially inhomogeneous condensate and its excitations (of an arbitrary nature). To achieve this, we introduced the so-called "form factor". This semi-phenomenological model of the Gross–Pitaevskii equation with a form factor allows to describe the physics of practically any quasiparticles arising in the considered quantum Bose gases.

The zoology of the form factors appearing in the GP equation is studied in detail. It is concluded that for a wide class of physical scenarios, the so-called quasilocal form factors are sufficient. We note that in the general case this form factor leads to a violation of translational invariance in the system, which is physically transparent: The distribution of the condensate is spatially nonuniform. Further, in the Thomas–Fermi approximation, a general solution of the GP equation with a quasilocal form factor is obtained. This solution has an interesting form in terms of a double rapidly convergent series, which can easily be shown by direct construction of the majorant of the corresponding series. Let us note some more beautiful properties of the obtained solution. First of all, it has a universal form with respect to the dimension of the space $D$ (for specific calculations we limited ourselves to three-dimensional Bose gases, the corresponding interactions in which were obtained in the classical theory). Moreover, the solution obtained universally includes all considered types of Bose gases.

To illustrate the results obtained, plots of condensate density functions for the exponential-trigonometric form factor are constructed. The exponent has good properties for numerical calculations and graphical constructions, and the periodic argument simulates the presence of a condensate density wave in the system. For the sake of completeness the GP equation with the optical lattice potential in the limit of small condensate concentrations is also considered in this paper. This limit does not distinguish between dipolar, quadrupolar and octupolar gases, and the equation itself is the well-known Schrödinger equation in the periodic potential (its stationary case is the Mathieu equation). The paper gives a brief discussion of the latter.

Then an important analysis of the stability of condensate was performed, in other words, a study of condensate excitations. In the Gaussian approximation, a functional describing the perturbations of the condensate is derived in detail from the Gross–Pitaevskii functional. We only consider the case of a quasilocal form factor of a special type. We note that this problem is a generalized analog of the Bogoliubov transformation used in the study of quantum Bose gases in operator formalism. In addition to the presence of a structure in the system, another generalization is that the probe wave function of the condensate perturbation does not have to be a plane wave, which, however, was chosen in this paper in order to obtain the spectrum of Bogoliubov excitations.

From the Bogoliubov spectrum, an equation describing the threshold perturbation momentum for the onset of the instability is obtained. This equation has an original form, since it includes the form factor of the theory in a complicated way. Another important result of our paper is that this equation makes it possible to establish the dependence of the threshold on the parameters of a stationary condensate. The latter is demonstrated by the example of the dependence of the threshold on the characteristic momentum of the condensate. For the sake of completeness, an approximating expression for the corresponding dependence is obtained in the paper. The approximating equation has the form of a certain rapidly converging series. The last statement is again easily proved by direct construction of the majorant. An interesting property of the equation obtained is that it has a universal form with respect to the dimension of the space $D$ (for specific calculations we again limited ourselves to three-dimensional Bose gases, the corresponding interactions in which were obtained

in the classical theory). The plots of the corresponding series for the exponential-trigonometric form factor are constructed.

In conclusion of the paper, let us note the question of the experimental determination of the form factor of the theory. What approach can give an appropriate hint? To answer this question, let us consider the hydrodynamics of quantum Bose gases of dipoles, quadrupoles, and octupoles. To this end, we represent the wave function of a Bose gas in the form:

$$\psi\left(t,\boldsymbol{r}\right) = \sqrt{n\left(t,\boldsymbol{r}\right)}\,e^{i\varphi(t,\boldsymbol{r})}. \tag{119}$$

The hydrodynamic velocity is given by:

$$\boldsymbol{V}\left(t,\boldsymbol{r}\right) = \frac{\hbar\partial_{\boldsymbol{r}}\varphi\left(t,\boldsymbol{r}\right)}{m}. \tag{120}$$

Then the continuity equation for condensate density is derived from the Gross–Pitaevskii equation:

$$\partial_t n\left(t,\boldsymbol{r}\right) + \partial_{\boldsymbol{r}}\left[n\left(t,\boldsymbol{r}\right)\boldsymbol{V}\left(t,\boldsymbol{r}\right)\right] = 0. \tag{121}$$

This equation is one of the two equations describing the hydrodynamic of ultracold gases. The second equation is the generalized Euler equation, which is given by:

$$m\partial_t \boldsymbol{V}\left(t,\boldsymbol{r}\right) = \partial_{\boldsymbol{r}}P_q\left(t,\boldsymbol{r}\right) - \partial_{\boldsymbol{r}}T\left(t,\boldsymbol{r}\right) - \partial_{\boldsymbol{r}}v\left(\boldsymbol{r}\right) - \int d^D\boldsymbol{r}'\,\partial_{\boldsymbol{r}}\Gamma\left(\boldsymbol{r},\boldsymbol{r}'\right)n\left(t,\boldsymbol{r}'\right), \tag{122}$$

where

$$P_q\left(t,\boldsymbol{r}\right) = \frac{\hbar^2}{2m}\frac{\partial_{\boldsymbol{r}}^2\sqrt{n\left(t,\boldsymbol{r}\right)}}{\sqrt{n\left(t,\boldsymbol{r}\right)}}, \quad T\left(t,\boldsymbol{r}\right) = \frac{mV^2\left(t,\boldsymbol{r}\right)}{2}. \tag{123}$$

Let us consider the last term of the Euler equation and do the substitution:

$$U\left(\boldsymbol{r}_1 - \boldsymbol{r}_2\right) \to g\delta^{(D)}\left(\boldsymbol{r}_1 - \boldsymbol{r}_2\right). \tag{124}$$

Then this term is given by:

$$\frac{1}{g}\int d^D\boldsymbol{r}'\,\partial_{\boldsymbol{r}}\Gamma\left(\boldsymbol{r},\boldsymbol{r}'\right)n\left(t,\boldsymbol{r}'\right) \to \partial_{\boldsymbol{r}}\left[\frac{n\left(t,\boldsymbol{r}\right)}{f^2\left(\boldsymbol{r}\right)}\right] = \frac{\partial_{\boldsymbol{r}}n\left(t,\boldsymbol{r}\right)}{f^2\left(\boldsymbol{r}\right)} - 2n\left(t,\boldsymbol{r}\right)\frac{\partial_{\boldsymbol{r}}f\left(\boldsymbol{r}\right)}{f^3\left(\boldsymbol{r}\right)}. \tag{125}$$

So we have obtained the equalities that answer the question. It follows from these equalities that the form factor can be determined experimentally by conducting experiments on the hydrodynamics of ultracold gases, since they explicitly determine the corresponding generalized Euler equation. Moreover, these equations are an alternative approach for determining, in particular, the Bogoliubov spectrum for system excitations. The latter is obtained by a well-known analysis of the stability of the system of equations of hydrodynamics.

Hydrodynamic equalities also show that quasilocal form factors are the most "natural": for the appearance of non-locality of the generalized Euler equation, more complex physical mechanisms may be required. For this reason, the phenomenological nature of the model is minimal. Moreover, even this model is not fully described in the paper: some calculations are performed for a quasilocal form factor of a special type. Calculations in the general case can be the subject of a separate publication. Moreover, a deeper study of the hydrodynamics and thermodynamics of quantum Bose gases in a model with a form factor deserves special attention, in particular, the consideration of temperature. Another interesting problem arises here: The modification of the form factor in the case of finite temperatures. Thus, the proposed description of quantum Bose gases in terms of the Gross–Pitaevskii equation with a form factor is interesting both from the theoretical point of view and for practical applications such as experiments with Bose gases of dipoles, axisymmetric quadrupoles and octupoles.

**Author Contributions:** A.A.A.: software, validation, formal analysis, investigation, writing—original draft preparation, visualization; A.U.B.: software, validation, formal analysis, investigation, writing—original draft preparation, visualization; S.L.O.: conceptualization, methodology, formal analysis, investigation, writing—original draft preparation, writing—review and editing, supervision. All authors have read and agreed to the published version of the manuscript.

**Funding:** This research received no external funding.

**Acknowledgments:** We are very grateful to our colleagues at the Department of Higher Mathematics MIPT and at the Department of Theoretical Physics MIPT for helpful discussions and comments. We express special gratitude to Sergey E. Kuratov and Alexander V. Andriyash for supporting this research at an early stage at the Center for Fundamental and Applied Research (Dukhov Research Institute of Automatics). Finally, we are very grateful to all MDPI Reviewers for many valuable comments and advice on this paper.

**Conflicts of Interest:** The authors declare no conflict of interest.

## Abbreviations

The following abbreviations are used in this manuscript:

BCS     Bardeen–Cooper–Schrieffer
BEC     Bose–Einstein Condensation
GP       Gross–Pitaevskii
QFT     Quantum Field Theory

## Appendix A

*Appendix A.1. Calculation of the k-Representation of the D-D Interaction*

In this appendix, detailed derivations of the interaction potentials of classical gases with different multipole values both in the coordinate $r$ and in the momentum $k$ representations are given. We begin with calculating the D-D interaction in the $k$-representation.

The expression for the D-D interaction energy in $r$-representation is given by:

$$U_D(r, \psi) = \frac{d^2(1 - 3\cos^2\psi)}{r^3}. \tag{A1}$$

In a general case the angle $\psi$ between the vector of the dipole moment $d$ and the vector $r$ is related to the angle between the vector $d$ and $k$ as follows:

$$\cos\psi = \sin\alpha\sin\theta\cos(\varphi - \gamma) + \cos\alpha\cos\theta. \tag{A2}$$

Indeed, it follows directly from the Figure A1. The projection on some unit vector $a$ is given by:

$$\frac{ar}{r} = \sin\alpha\cos\gamma\sin\theta\cos\varphi + \sin\alpha\sin\gamma\sin\theta\sin\varphi + \cos\alpha\cos\theta. \tag{A3}$$

From this expression after the transformations (taking into account the spherical coordinate system) the connection between $\psi$ and $\alpha$ is obtained. From this formula it is clear that one may put $\gamma = 0$, because in the course of calculation of the $k$-representation, one needs to integrate the expression with $\cos(\varphi - \gamma)$ over the period $2\pi$.

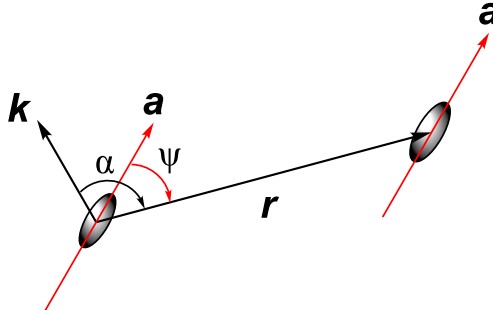

**Figure A1.** The relation between the angles between the axis of symmetry of the multipole (the unit vector *a*) and the vectors *r* and *k*

The integral that has to be calculated is given by:

$$\int_0^\infty dr \int_0^\pi d\theta \int_0^{2\pi} d\varphi \, e^{-ikr\cos\theta} \sin\theta \frac{1 - 3\cos^2\psi}{r}. \tag{A4}$$

After the integration over $\varphi$ and the substitution $x = \cos\theta$ with further integration over $x$ in corresponding limits, we see that the integral has a divergence at zero when integrating with respect to $r$. Thus we integrate beginning from a certain $r_0$. This leads to the expression:

$$-\frac{2\pi(1 + 3\cos 2\alpha)(r_0 k \cos r_0 k - \sin r_0 k)}{r_0^3 k^3}, \tag{A5}$$

which in the limit of $r_0 \to 0$ gives

$$U_D(\boldsymbol{k}, \alpha) = \frac{4\pi d^2}{3}(3\cos^2\alpha - 1). \tag{A6}$$

It is clear that the expression does not depend on *k*.

*Appendix A.2. Calculation of the k-Representation of the Q-Q Interaction*

In a general case the angle $\psi$ between the quadrupole axis and the vector *r* is related to the angle $\alpha$ between the vector *d* and *k* by the Equation (A2). $\alpha$ is the angle between the quadrupole axis and the vector *k*.

The integral that has to be calculated is given by:

$$\int_0^\infty dr \int_0^\pi d\theta \int_0^{2\pi} d\varphi \, r^2 \sin\theta \, U_D(\boldsymbol{r}, \alpha)e^{-i\boldsymbol{kr}}. \tag{A7}$$

After integration over $\varphi$:

$$\int_{\theta, r} r^2 \sin\theta \, U_Q(\boldsymbol{r}, \alpha)e^{-i\boldsymbol{kr}} \frac{\pi(9 + 20\cos 2\alpha + 35\cos 4\alpha)}{256}(9 + 20\cos 2\theta + 35\cos 4\theta), \tag{A8}$$

where $\int_{\theta, r} \equiv \int d\theta \int dr \sin\theta r^2$. In order to calculate the integral over $\theta$ one has to do the substitution $x = \cos\theta$. Finally, after integrating over *R*:

$$U_Q(\boldsymbol{k}, \alpha) = \frac{\pi A_0 k^2}{210}(9 + 20\cos 2\alpha + 35\cos 4\alpha), \tag{A9}$$

which after calculations gives (here $A_0 = 3Q^2/16$):

$$U_Q(\boldsymbol{k}, \alpha) = \frac{3Q^2 \pi k^2}{16}\left(\frac{4}{35} - \frac{8}{7}\cos^2\alpha + \frac{4}{3}\cos^4\alpha\right). \tag{A10}$$

This is the expression for the Q-Q interaction energy in the **k**-representation.

*Appendix A.3. Calculation of the r-Representation of the O-O Interaction*

The calculation of the interaction has no conceptual complexity, but it is very cumbersome, even if taking into account the symmetry of the considered case. We will describe the general idea of calculations, and the details can be implemented by using any of computer algebra systems and programming languages.

Calculations begin with an expression for the potential created by the quadrupole:

$$\varphi(\boldsymbol{r}) = O_{\lambda\nu\mu}\frac{r_\lambda r_\nu r_\mu}{r^4}. \tag{A11}$$

Next, let us remind that the expression for the energy of the O-O interaction in the **r**-representation is given by:

$$U_O(\boldsymbol{r}, \psi) = \frac{O_{\alpha\beta\gamma}O_{\lambda\mu\nu}}{540}\partial_\alpha\partial_\beta\partial_\gamma\left(\frac{r_\lambda r_\nu r_\mu}{r^4}\right). \tag{A12}$$

Thus, the following equation needs to be calculated first:

$$\partial_\alpha\partial_\beta\partial_\gamma\left(\frac{r_\lambda r_\nu r_\mu}{r^4}\right). \tag{A13}$$

Then, we should use the fact that the octupole moment tensor has only 7 independent components and symmetry properties which were mentioned earlier. This lets us separate the resulting terms into groups, and then individually simplify each group. Finally, it is necessary to substitute the spherical coordinates into the resulting expression and simplify it.

*Appendix A.4. Calculation of the k-Representation of the O-O Interaction*

To calculate the O-O interaction in momentum space, one must substitute the expression for the angles $\psi$ and $\alpha$. After this, the problem simplifies to calculating the integral:

$$\int_0^\infty dr \int_0^\pi d\theta \int_0^{2\pi} d\varphi\, r^2 e^{-ikr\cos\theta}\sin\theta\, U_O(\boldsymbol{r}, \alpha). \tag{A14}$$

Integration over $\phi$ is obvious, integration over $\theta$ is simplified by the substitution $x = \cos\theta$. When integrating over $r$, there are no problems with divergence. The whole process is easy to do in any suitable system of computer algebra, there is no point in writing the intermediate calculations because of their cumbersomeness.

The final result is given by:

$$U_O(\boldsymbol{k}, \alpha) = \pi k^4 O^2 \left(\frac{5\cos^2\alpha}{3564} - \frac{5\cos^4\alpha}{1188} + \frac{\cos^6\alpha}{324} - \frac{5}{74844}\right). \tag{A15}$$

After simplifications one gets the expression given in the corresponding section.

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
