# Peer review of "Quantum Gases of Dipoles, Quadrupoles and Octupoles in Gross–Pitaevskii Formalism with Form Factor"

_condensedmatter, doi:10.3390/condmat5040061_

Round 1

Reviewer 1 Report

The authors consider a somewhat exotic generalization of the standard theory of quantum gases, such as Bose-Einstein condensates, including form factors and multipole interactions among other things. While I am not sure about the physical motivation and potential impact of this approach, the article is clear and well-written and the mathematical analysis seems sound. Results are discussed in detail. 

In summary, I cannot reject the publication of this paper and I hope that it might be of interest for some researchers.

Author Response

Dear Reviewers, we are of a great appreciation for your Reports on our manuscript. In accordance with these Reports, we have made corrections in our paper.

Into the new version of the paper we have added your comments using additional notes (stickers) as well as we highlighted in green all the corrections that we made to the paper. The new paper file is “Quantum Gases MDPI (Condensed Matter)”. Please find your comments and our answers in this file. We assumed that the highlighting of responses in the new version of the paper will be convenient for you. If a line-by-line listing of changes is required, please let us know about it.

We sincerely hope that we have given detailed answers to all your comments.

Yours sincerely,
Stanislav L. Ogarkov and co-authors

Reviewer 2 Report

The manuscript deals with unusual, but interesting physics of quantum gases. On the hand dipole, quadrupole and octupole interactions in the mean-field Gross–Pitaevskii are considered. On the other the focus is on the impact of form factors. In both cases, however, the major deficiency of the manuscript is the lack of any connection to experiments. Thus, the authors should review

1) In which context appear apart from dipolar interactions also quadrupole and octupole interactions in the realm of ultracold quantum gases? It would be preferable if you could mention explicitly examples of quantum gases with their respective dipole, quadrupole and octupole interaction strength.

2) Where do form factors play an experimental role? In which experiments are they measured and how they are modelled in order to describe the measurements.

Furthermore, the manuscript contains partially superfluous theoretical material, which distracts the reader from the main message. One example is the excursion into quantum field theory in the introduction. Here various correct statements are made, but their context to the different types of interactions and form factors, respectively, is not given. And a third major criticism is that the English language of the manuscript should be polished.

Thus, in conclusion, the manuscript needs a severe revision along the line „less theory – more experiment“ in order to underline the motivation for the work.

Author Response

(The authors gave the same response as above.)

Round 2

Reviewer 2 Report

The authors have revised their manuscript according to the suggestions of the referees. Therefore, the manuscript can now be accepted for publication.